# Time-resolved measurements of the densities of individual frozen hydrometeors and fresh snowfall

**Dhiraj K. Singh**[1]**, Eric R. Pardyjak**[1]**, and Timothy J. Garrett**[2]

[1]Department of Mechanical Engineering, University of Utah, Salt Lake City, UT, USA
[2]Department of Atmospheric Sciences, University of Utah, Salt Lake City, UT, USA

**Correspondence:** Eric R. Pardyjak (pardyjak@eng.utah.edu)

**Abstract.** It is a challenge to obtain accurate measurements of the microphysical properties of delicate, structurally complex, frozen, and semi-frozen hydrometeors. We present a new technique for the real-time measurement of the density of freshly fallen individual snowflakes. A new thermal-imaging instrument, the Differential Emissivity Imaging Disdrometer (DEID), has been shown through laboratory and field experiments to be capable of providing accurate estimates of individual snowflake and bulk snow hydrometeor density (which can be interpreted as the snow-to-liquid ratio or SLR). The method exploits the rate of heat transfer during the melting of a hydrometeor on a heated metal plate, which is a function of the temperature difference between the hotplate surface and the top of the hydrometeor. The product of the melting speed and melting time yields an effective particle thickness normal to the hotplate surface, which can then be used in combination with the particle mass and area on the plate to determine a particle density. Uncertainties in estimates of particle density are approximately 4 % based on calibrations with laboratory-produced particles made from water and frozen solutions of salt and water and field comparisons with both high-resolution imagery of falling snow and traditional snowpack density measurements obtained at 12 h intervals. For 17 storms, individual particle densities vary from 19 to 495 kg m$^{-3}$, and storm mean snow densities vary from 40 to 100 kg m$^{-3}$. We observe probability distribution functions for hydrometeor density that are nearly Gaussian with kurtosis of $\approx 3$ and skewness of $\approx 0.01$.

## 1 Introduction

Frozen and semi-frozen hydrometeors have a very wide range of porosities (Dunnavan et al., 2019). Determining their particulate densities and bulk snow-to-liquid ratios (SLRs) once fallen on the ground is important to a wide range of fields, including hydrology (Rango and Martinec, 1995; Sturm et al., 2010), climatology (Dickinson, 1983), remote sensing at wavelengths ranging from the visible to the microwave (Kendra et al., 1994; Kokhanovsky and Zege, 2004; Gergely et al., 2010), and the parameterization of snowflake fall speeds in weather and climate models (Rutledge and Hobbs, 1984; Hong et al., 2004; Fovell and Su, 2007; Alcott and Steenburgh, 2010; Finlon et al., 2019). Because hydrometeor porosity is invisible to most imaging techniques, obtaining accurate snowflake density estimates has proven to be a significant challenge where even the best estimates have required the use of sophisticated field programs using multiple instruments (Tiira et al., 2016).

Improvements to weather prediction are currently hampered by an inability to assimilate information about ongoing variability in frozen and semi-frozen precipitation particles (Rasmussen et al., 2011). Avalanche forecasting in mountainous regions depends, in part, on knowledge of the vertical density structure of freshly fallen snow (Morrison et al., 2023), a parameter that is typically measured at sparse intervals (Schweizer et al., 2011; Proksch et al., 2016) using techniques such as micro-computed tomography ($\mu$CT) or, more typically, manual gravimetric methods (Proksch et al., 2016).

In our previous work, we showed that a new thermal-imaging instrument, the Differential Emissivity Imaging Dis-

drometer (DEID), can be used to measure individual hydrometeor density based on the first automated direct measurements of particle mass in combination with estimates of the spherical-particle-equivalent effective diameter or by using concurrent photographic imagery of the morphological characteristics of hydrometeors as they fall (Singh et al., 2021; Rees et al., 2021). While the spherical-particle approach offers the advantage of simplicity, it was found to lead to snowflake density estimates that were significantly biased low relative to a method that required an added camera system, likely because snowflakes are not in fact spheres. Here, we describe a new method for estimating particle-by-particle frozen-hydrometeor density that, like the spherical-particle method, uses only the DEID to measure mass but instead infers particle volume from DEID measurements of melting time and particle area and estimates of the rate of heat transfer from the hotplate to the hydrometeor to obtain a melting speed (MS).

## 2 DEID measurement techniques for obtaining hydrometeor mass and density

The DEID consists of an infrared camera pointed at the surface of a low-emissivity aluminum hotplate. To quantify hydrometeor area on the hotplate, the DEID makes use of the contrasting thermal emissivities of water ($\varepsilon > 0.95$) and aluminum ($\varepsilon < 0.1$) at the same temperature. Owing to the high difference in emissivity, melted hydrometeors with nearly the same thermodynamic temperature as the heated plate have strongly contrasting radiative temperatures such that droplets on the heated plate can be easily discriminated using a thermal camera. The hotplate surface is roughened, which prevents displacement of melted snowflakes at high wind speeds, as demonstrated in wind-tunnel experiments with wind speeds varying from 2 to $12\,\text{m}\,\text{s}^{-1}$.

### 2.1 Particle mass measurement

The DEID methodology for obtaining the mass of a hydrometeor particle has previously been described by Singh et al. (2021), Rees et al. (2021), Rees and Garrett (2021), and Morrison et al. (2023). Here, we present a concise summary including recent modifications to the measurement methodology. Briefly, the mass of individual hydrometeors is obtained by considering the contact area of each hydrometeor on a heated metal plate and the temperature difference between the plate and the surface of the melted liquid particle, which is integrated over time from the point of first impact of the particle onto the plate surface up to the point of its complete evaporation.

Specifically, the mass $m$ of an individual hydrometeor is obtained by applying conservation of energy to a control volume surrounding each hydrometeor on the hotplate (see Fig. 1). The heat gained by a snowflake from the heated plate

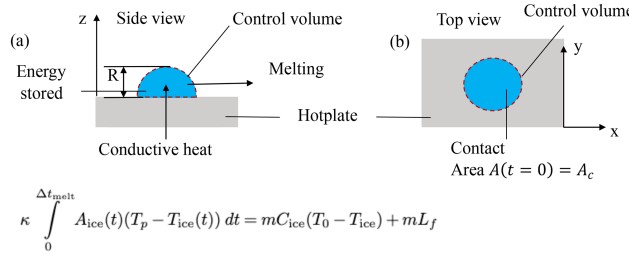

$$\kappa \int_0^{\Delta t_\text{melt}} A_\text{ice}(t)(T_p - T_\text{ice}(t))\,dt = mC_\text{ice}(T_0 - T_\text{ice}) + mL_f$$

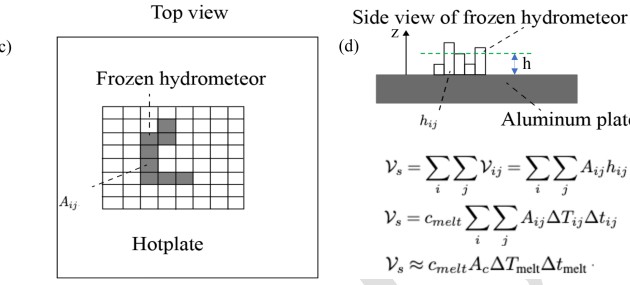

$$\mathcal{V}_s = \sum_i \sum_j \mathcal{V}_{ij} = \sum_i \sum_j A_{ij} h_{ij}$$

$$\mathcal{V}_s = c_\text{melt} \sum_i \sum_j A_{ij} \Delta T_{ij} \Delta t_{ij}$$

$$\mathcal{V}_s \approx c_\text{melt} A_c \Delta T_\text{melt} \Delta t_\text{melt}\,.$$

**Figure 1.** For the methodology presented here, a control volume is defined to wrap around the hydrometeor. **(a)** Side view of control volume. Schematic of the heat-transfer process from the DEID's aluminum hotplate to a solid hydrometeor along the $z$ axis during melting. $R$ is the radius of the hemispherical ice particle on the hotplate, $h$ is the effective thickness of the hemisphere that is $2R/3$, and $m$ is the mass of the ice particle. **(b)** Top view of the control volume. $A_c$ is the contact area of the ice particle in the $x-y$ plane. **(c)** Schematic illustrating the method for calculating the contact area of a frozen hydrometeor with an arbitrary geometry on the hotplate. The individual rectangles represent pixels as viewed with the thermal camera. **(d)** Schematic illustrating the method for calculating the volume of a hydrometeor using the contact area and height of a frozen hydrometeor with an arbitrary geometry on the hotplate. $h = (1/N)\sum_i \sum_j h_{ij}$, where $h_{ij}$ is the height of a frozen hydrometeor normal to the hotplate that is associated with the $ij$th pixel at $t = 0$, and $N$ is the total number of pixels associated with a frozen hydrometer at $t = 0$.

is assumed to be equivalent to the heat required to increase the snowflake's internal energy and the heat lost during melting and evaporation, which may be described as follows:

$$Q_\text{in} - Q_\text{out} = \Delta Q_\text{st}. \tag{1}$$

Here, $Q_\text{in}$ is the conductive heat gained by a hydrometer from the hotplate, $Q_\text{out}$ is the heat loss from a hydrometeor to the surroundings, and $\Delta Q_\text{st}$ is the energy stored in a hydrometer during melting and evaporation. Neglecting convection and radiation losses from the hydrometer (which was shown to be a good approximation in Singh et al., 2021), Eq. (1) can be written as

$$\int_{t_\text{pc}}\int_{A_c} k(\partial T/\partial z)\mathrm{d}A_c\,\mathrm{d}t - \int_\mathcal{V} \rho L_\text{eqv}\mathrm{d}\mathcal{V}$$

$$= \int_{t_\text{pc}}\int_\mathcal{V} \rho C_p (\partial T/\partial t)d\mathcal{V}\mathrm{d}t. \tag{2}$$

Here, $k$ is the thermal conductivity of the aluminum plate, $A_c$ is the contact area of the hydrometeor on the hotplate, $t_{pc}$ is the time it takes to melt or evaporate a droplet (phase change), $\partial T/\partial z$ is the temperature gradient related to conduction from the hotplate into the hydrometeor, $\rho$ is the density of the hydrometeor, $\mathcal{V}$ is the volume of the hydrometeor, $L_{eqv} = L_f + L_v$ is the combined latent heat of fusion ($L_f$) and vaporization ($L_v$), $C_p$ is the specific heat of the hydrometeor, and $z$ is the vertical direction normal to the hotplate. Using $m = \int_{\mathcal{V}} \rho \, d\mathcal{V}$, $\partial T/\partial z \approx \Delta T(x, y, t)/\Delta z$ and by separating the energy stored in the hydrometeor during melting and evaporation, Eq. (2) can be modified as

$$\int_{t_{pc}} \int_{A_c} (k/\Delta z)_{eff} \Delta T(x, y, t) \mathrm{d}A_c \, \mathrm{d}t - m(L_f + L_v)$$
$$= mC_{ice}(T_0 - T_{ice}) + mC_{liq}(T_p - T_0). \tag{3}$$

Here, $C_{ice}$ is the specific heat of ice; $T_p$ is the hotplate surface temperature during melting and evaporation (it is constant with time); $T_{ice}$ is the initial temperature of a frozen hydrometer; $T_0 = 0\,°C$; $L_f = 3.34 \times 10^5\,\mathrm{J\,kg^{-1}}$ is the latent heat of fusion of water; $L_v = 2.32 \times 10^6\,\mathrm{J\,kg^{-1}}$ is the latent heat of vaporization of water (see Appendix A1); $C_{liq} = 4.18 \times 10^3\,\mathrm{J\,kg^{-1}\,K^{-1}}$ is the specific heat of liquid water; and $(k/\Delta z)_{eff} = \kappa$ is an empirical device-specific calibration coefficient related to the amount of heat that passes through the metal plate into individual hydrometeors per unit of time through a unit area with a temperature gradient of $1\,°$, determined to be $7.01 \pm 0.01 \times 10^3\,\mathrm{W\,m^{-2}\,K^{-1}}$ (see Singh et al., 2021, for details), which is independent of particle size and environmental conditions. In practice, to numerically calculate the mass of a hydrometeor using images from a thermal camera on a "pixel-by-pixel" basis, we suppose that $n_x$ is the total number of pixels making up the hydrometeor in the $x$ direction at time $t$, and $n_y$ is the total number of pixels associated with the hydrometeor in the $y$ direction at time $t$. The total number of pixels at time $t$ is $N(t) = n_x(t)n_y(t)$. Equation (3) can be modified and rearranged for a pixel-by-pixel implementation of Eq. (3) as

$$mC_{ice}(T_0 - T_{ice}) + mL_f + mC_{liq}(T_p - T_0) + mL_v$$
$$= \kappa \int_0^{\Delta t_{evap}} \sum_i^{n_x} \sum_j^{n_y} (T_p - T_{ij}(t)) A_{ij}(t) \, \mathrm{d}t. \tag{4}$$

Here, $\Delta t_{evap}$ is the time required to melt and evaporate a hydrometeor, $A_{ij}(t)$ is the area of the $ij$th pixel at time $t$, and $T_{ij}(t)$ is the temperature of the $ij$th pixel at time $t$. Assuming all pixels have the same area (e.g., $A_1$), then $\sum_i^{n_x} \sum_j^{n_y} A_{ij}(t) = n_x n_y A_1(t)$ is the total contact area of the hydrometeor at time $t$ (i.e., $n_x n_y A_1(t) = A_c(t)$). Substituting $A_c(t)$ into Eq. (4) yields

$$mC_{ice}(T_0 - T_{ice}) + mL_f + mC_{liq}(T_p - T_0) + mL_v$$
$$= \kappa \int_0^{\Delta t_{evap}} A_c(t)(T_p - \frac{1}{n_x n_y} \sum_i^{n_x} \sum_i^{n_x} T_{ij}(t)) \, \mathrm{d}t. \tag{5}$$

Here, $\frac{1}{n_x n_y} \sum_i^{n_x} \sum_i^{n_x} T_{ij}(t) = T_h(t)$. $T_h(t)$ is the spatial mean temperature over all frozen hydrometeors and/or water droplet pixels at time $t$ on the hotplate. Substituting the spatial mean temperature $T_h(t)$ into Eq. (5) yields

$$mC_{ice}(T_0 - T_{ice}) + mL_f + mC_{liq}(T_p - T_0) + mL_v$$
$$= \kappa \int_0^{\Delta t_{evap}} A_c(t) (T_p - T_h(t)) \, \mathrm{d}t. \tag{6}$$

In Eq. (6), at any time $t$ during melting and evaporation, the temperature of some portions of the hydrometeor area is less than 0 and some temperatures are greater than 0. The area of a hydrometer at any time $t$ can be written as $A_c(t) = A_{ice}(t) + A_{liq}(t)$, where $A_{ice}(t)$ is the contact area of the hydrometeor fraction on the hotplate with a temperature less than or equal to 0 at time $t$ (the sum of all pixels with temperatures less than or equal to 0), and $A_{liq}(t)$ is the contact area of hydrometeors on the hotplate with a temperature greater than 0 at time $t$ (sum of all pixels with a temperature greater than 0). After substituting $A(t)$ into Eq. (6), Eq. (6) may be re-written as

$$mC_{ice}(T_0 - T_{ice}) + mL_f + mC_{liq}(T_p - T_0) + mL_v$$
$$= \kappa \int_0^{\Delta t_{melt}} (T_p - T_{ice}(t)) A_{ice}(t) \, \mathrm{d}t$$
$$+ \kappa \int_{t_0}^{\Delta t_{evap}} (T_p - T_{liq}(t)) A_{liq}(t) \, \mathrm{d}t. \tag{7}$$

Here, $t = 0$ corresponds to when a frozen hydrometeor hits the hotplate, and $t = t_0$ is when the thermal camera sees the liquid portion for the first time. $t_0$ is the time lag between the melting and evaporation start times, which is about $0.1\,\mathrm{s}$ for the laboratory ice particles tested and negligible for snowflakes observed using our typical thermal-camera recording frame rates. $T_{ice}(t)$ is the spatial mean temperature of all pixels with a temperature less than or equal to 0, and $T_{liq}(t)$ is the spatial mean temperature of all pixels with a temperature greater than 0. Note that in post-processing data, the thermal camera can be adjusted to selectively "see" particles on the hotplate in specific temperature ranges (see Sect. 3 and Fig. E1). To more easily evaluate Eq. (7), the camera is set to only see hydrometeors on the hotplate after melting (details are given in Fig. E1). We assume that heat transfer through $A_{ice}$ exclusively leads to increases in ice temperature and melting, and heat transfer through $A_{liq}$ does not go

into these ice portions. We justify this since the temperature gradients between the ice and water are much smaller than the temperature gradients between the plate and hydrometeor. Furthermore, the thermal conductivity of aluminum is much higher than that of ice or water. Hence, $A_{ice}(t) = 0$ for $T_h(t) < 0\,°C$ and $\kappa \int_0^{\Delta t_{melt}} (T_p - T_h(t))A_{ice}(t)dt = 0$, which is equal to $mC_{ice}(T_0 - T_{ice}) + mL_f$. This trick allows us to remove both terms from Eq. (7), which yields

$$m = \frac{\kappa}{C_{liq}T_p + L_v} \int_{t_0}^{\Delta t_{evap}} (T_p - T_{liq}(t))\,A_{liq}(t)\,dt. \tag{8}$$

Here, the lower integration bound is reset so that $t_0 = 0$ such that Eq. (8) is evaluated for the liquid phase only, and the time integral is evaluated using the trapezoidal rule. $T_{liq}(t)$ is the spatial mean water droplet temperature at time $t$. Note that the initial and final temperatures of all liquid droplets during evaporation are $T_0 = 0\,°C$ and $T_p$, respectively, and more than $\approx 98\,\%$ of the mass evaporates at the highest temperature ($\approx T_p$). This is illustrated in the time series of the temperature of an evaporating hydrometeor given in Fig. A1. Note that we use the subscripts ice and s to denote frozen hydrometeors (i.e., ice and snowflake, respectively).

Mass estimates were shown in wind-tunnel calibrations to be nearly independent of environmental conditions, including wind speed, relative humidity, and ambient temperature (Singh et al., 2021). Specifically, wind-tunnel experiments with the DEID showed less than 4 % variability in mass measurements of hydrometeors for a wide range of wind speeds, relative humidities, and air temperatures (Singh et al., 2021). The reason for the low sensitivity to environmental conditions is that the DEID directly measures the energy required to melt and evaporate a droplet, $mL_{eqv}$. For example, the heat-transfer rate to a droplet is dependent on parameters such as wind speed through the Reynolds number (Kosky et al., 2013) and the temperature. However, while higher winds may accelerate heat transfer, they also diminish the time for completing evaporation. Because the product of the heat-transfer rate and evaporation time determines particle mass, winds play a minor role in the calculation of mass.

## 2.2 Particle density

Obtaining frozen-hydrometeor density from hydrometeor mass requires an estimate of the particle volume while it is in its frozen state. While the DEID can provide an accurate estimate of snowflake mass $m$ and initial snowflake contact area after it impacts the hotplate $A_c$, it cannot provide a direct measure of a particle's effective thickness in the direction normal to the hotplate $h$ as illustrated in Fig. 1. In its place, we have developed a method for estimating $h$ based on a "melting speed" $v_{melt}$ such that $h = v_{melt}\Delta t_{melt}$, where $\Delta t_{melt}$ is the time required to melt an individual snowflake. Using these substitutions, the density of a frozen hydrometeor can be written as

$$\rho_{MS} = \frac{m}{\mathcal{V}_s} = \frac{m}{A_c v_{melt}\Delta t_{melt}}, \tag{9}$$

where $\rho_{MS}$ indicates the density computed using the melting speed method, and $\mathcal{V}_s$ is the volume of a snowflake estimated as $A_c v_{melt}\Delta t_{melt}$.

We propose a method to measure $v_{melt}$ as a function of the temperature difference across a hydrometeor ($\Delta T_{melt}$) – and hence the heat-transfer rate – as illustrated in Fig. 1. During the time it takes to melt a snowflake, $\Delta t_{melt}$, a hydrometeor receives a quantity of energy equal to $mL_{ff} = mC_{ice}(T_0 - T_{ice}) + mL_f$, i.e., the sum of the internal energy of a frozen hydrometeor and its latent heat of fusion (energy received by the ice to increase temperature and melt completely) from the hotplate, which is independent of the density of a frozen hydrometeor. $v_{melt}$ is associated with the conductive heat flux ($\kappa\Delta T_{melt}$) from the hotplate to the frozen hydrometeors during melting. We hypothesize that $v_{melt}$ is a function of the temperature difference across a hydrometeor ($\Delta T_{melt}$), and we may write $v_{melt}$ as

$$v_{melt} = c_{melt}\Delta T_{melt}, \tag{10}$$

where $\Delta T_{melt} = \overline{T_p - T_s(t)}$ (the overbar represents a temporal mean) during the melting process, $T_s(t)$ is the spatial mean of the surface temperature of the frozen portion of the particle during melting, and $c_{melt}$ is a constant determined experimentally (see Sect. 4.1).

Now, if $v_{melt}$ from Eq. (10) is substituted into Eq. (9), the MS density equation can now be written as

$$\rho_{MS} = \frac{m}{c_{melt}A_c\Delta T_{melt}\Delta t_{melt}}. \tag{11}$$

The melting parameter $\Delta t_{melt}$ is quite short for low-density snowflakes and hence requires high-frequency recording of thermal images, resulting in the generation of a tremendous amount of data, which is not convenient for field experiments. In field experiments, the DEID measures $\Delta t_{evap}$, which is much longer than $\Delta t_{melt}$. Fortunately, a relation between $\Delta t_{melt}$ and $\Delta t_{evap}$ can be derived easily (see Appendix B). By estimating the average conductive heat-transfer rate during the melting and evaporation processes, we may substitute $\Delta T_{melt}\Delta t_{melt} \approx (L_{ff}/L_{vv})(\Delta T_{evap}\Delta t_{evap})$ into Eq. (11), which yields

$$\rho_{MS} = c\frac{m}{A_c\Delta T_{evap}\Delta t_{evap}}. \tag{12}$$

Here, $\Delta T_{evap} = \overline{T_p - T_{liq}(t)}$ during the evaporation process, and the constant, $c$, is given by

$$c = (L_{vv})/(L_{ff}c_{melt}) \tag{13}$$

in units of kelvin per second per meter ($K\,s\,m^{-1}$). The constant, $c$, is derived from a combination of thermodynamic and

laboratory calibration constants that must be determined experimentally (see Sect. 4.1).

In practice, Eq. (11) is evaluated following the methodology shown in Fig. 1b. Snowflakes with complex shapes do not have simple height relationships like $h = 2R/3$, as shown in Fig. 1a. Hence, a method is required to determine the height and volume of each pixel within a snowflake. If $A_{ij}$ is taken as the area of the $ij$th pixel at $t = 0$ and $h_{ij}$ is assumed to be the height of a frozen hydrometeor normal to the hotplate that is associated with the $ij$th pixel at $t = 0$, the effective thickness of frozen hydrometer $h$ can be written as $h = (1/N)\sum_i \sum_j h_{ij}$. Here, $N$ is the total number of pixels associated with a frozen hydrometeor; $h_{ij}$ can be estimated using $v_{melt}\Delta t_{ij}$, where $v_{melt} = c_{melt\Delta T_{ij}}$; and $\mathcal{V}_{ij}$ is the volume of the $ij$th pixel. We may then write Eq. (14) as

$$\mathcal{V}_s = \sum_i \sum_j \mathcal{V}_{ij} = \sum_i \sum_j A_{ij} h_{ij}$$
$$= c_{melt} \sum_i \sum_j A_{ij} \Delta T_{ij} \Delta t_{ij}, \tag{14}$$

which can subsequently be estimated using the MS method on a pixel-by-pixel basis. Here, $\Delta T_{ij}$ is the temporal mean temperature difference across the $ij$th pixel, $\Delta t_{ij}$ is the melting time of the $ij$th pixel, and $c_{melt}$ is the calibration constant of the melting velocity. This study used Eq. (14) to calibrate laboratory ice particles and compare snowflake habits. For field observations, we make the following simplification to determine the volume:

$$\mathcal{V}_s = c_{melt} A_c \frac{1}{N} \sum_i \sum_j \Delta T_{ij} \Delta t_{ij}$$
$$\approx c_{melt} A_c \Delta t_{melt} \frac{1}{N} \sum_i \sum_j \Delta T_{ij}$$
$$= c_{melt} A_c \Delta T_{melt} \Delta t_{melt} = \frac{1}{c} A_c \Delta T_{evap} \Delta t_{evap}. \tag{15}$$

Here, $A_c = N A_{ij}$ (assuming all pixels have the same area), and $\Delta T_{melt} = \frac{1}{N} \sum_i \sum_j \Delta T_{ij}$ is the spatial and temporal mean temperature difference across a frozen hydrometer during melting. Based on experimental tests, we assume $\Delta t_{melt} \approx \Delta t_{ij}$, which is the melting time of a frozen hydrometer, and $c = (L_{ff})/(L_{vv} c_{melt})$ [TS1] (see Appendix B).

Note that $\Delta t_{melt}$ is impacted by variability in environmental conditions. A sample time series of temperature and hydrometeor area during melting and evaporation is shown in Fig. 2. During the melting process, the area of a particle that is in its frozen state decreases to 0 from a maximum immediately after having fallen on the plate. At the same time, the observed liquid component of the hydrometeor increases to a maximum before abruptly disappearing. The sum of these two areas is nearly constant, at least accounting for inevitable uncertainties in the binary thresholding associated with discriminating the hydrometeor from its background. Notably, the sum is also equal to the initial area of the frozen particle

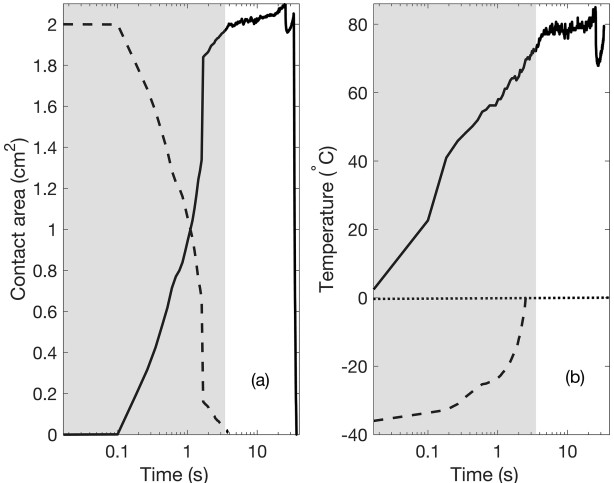

**Figure 2.** Observed melting and evaporation of a laboratory-made ice particle on the DEID hotplate. **(a)** Time series of the area of the ice particle, $A_{ice}(t)$ (dashed black line), and the liquid, $A_{liq}(t)$ (solid black line). **(b)** The minimum surface temperature of the ice particle (dashed black line) and the temperature of the liquid, $T_{liq}(t)$ (solid black line), immediately after being placed on the hotplate. The horizontal dotted line represents a temperature partition in the melting and evaporation process. The shaded region denotes the period of melting. Details on the manufacturer of the laboratory-made ice particles and experiments are provided in Sect. 3.1. The lag between the melting and evaporation start times is 0.1 s, which is equal to the thermal-camera sampling period (sampling rate of 10 Hz).

prior to its melting. Figure 2 shows how the area of the ice particle prior to melting is similar to the maximum area of the liquid droplet, showing that the area of solid hydrometeors is preserved after melting.

## 2.3 Use of the DEID to determine bulk-snowpack-derived quantities

In additional to individual hydrometeor mass and density measurements, the DEID can be used to provide useful bulk snowpack quantities. Precipitation intensity or snow water equivalent rate of precipitation, $\dot{SWE}$ (in $mm\,h^{-1}$), can be estimated from the cumulative particle mass measured by the DEID over a given time period ($\Delta t_{res}$) as

$$\dot{SWE} = k \frac{\Delta m}{\rho_w A_{hp} \Delta t_{res}}, \tag{16}$$

where $k$ is a conversion factor from meters per second to millimeters per hour ($3.6 \times 10^6\,mm\,h^{-1}\,m^{-1}\,s$); $\Delta t_{res}$ is the sampling time (h); $\Delta m$ (kg) is the total hydrometeor mass that falls on the hotplate at a given time, where the individual mass of the hydrometeor is estimated using Eq. (8); $\rho_w$ ($kg\,m^{-3}$) is the bulk density of water; and $A_{hp}$ ($m^2$) is a rectangular sampling area on the hotplate that captures all hydrometeors. The accumulated SWE (mm) can be calculated over a given time interval $\Delta t_{res}$ (h) as $SWE = \dot{SWE} \times \Delta t_{res}$.

The average density ($\overline{\rho}_{MS}$) of the snowflakes can be estimated using the DEID from the ratio of the cumulative measured mass and the total volume of all snowflakes sampled in a given time interval ($\Delta t_{res}$),

$$\overline{\rho}_{MS} = \frac{\sum_{i=1}^{N} m_i}{\sum_{i=1}^{N} m_i / \rho_{MS,i}}, \tag{17}$$

where $m_i$ (kg) is the mass of the $i$th snowflake, $\rho_{MS,i}$ (kg m$^{-3}$) is the density of the $i$th snowflake, and $N$ is the total number of snowflakes collected on the plate during the given time interval ($\Delta t_{res}$). Using the average density of the snowflakes during a specific time period and assuming no space between snowflakes or overlapping snowflakes, the new snow accumulation rate $\dot{H}$ (mm h$^{-1}$) is then

$$\dot{H} = k \frac{\Delta m}{\overline{\rho}_{MS} A_{hp} \Delta t_{res}}. \tag{18}$$

Finally, the total accumulated snow over $H$ (mm) over a given time interval $\Delta t_{res}$ (h) is given by $H = \dot{H} \times \Delta t_{res}$. Note that the bulk density of a fresh snowpack and the height of snowpack can differ from the average density of individual snowflakes ($\overline{\rho}_{MS}$) and $H$, respectively, because snowflake settling and compaction on the ground depend on considerations such as their settling characteristics, fall angle, wind speed, the structure of snowflakes, and ambient temperature. We do not account for these processes in the calculation of the volume of freshly fallen snow layers as the impacts are largely unknown. Hence, these variables are proxies for those in the actual snowpack.

## 3 Experimental methods

Two laboratory experiments and one field experiment were designed to calibrate and validate the MS method for determining snowflake density. The first lab experiment was used to estimate $v_{melt}$ of ice particles for a given set of environmental conditions and validate the density measurements of ice particles. The second lab experiment investigated the impact of environmental factors on $v_{melt}$. A field experiment was conducted at Alta Ski Area's mid-Collins snow-study plot to provide an opportunity to validate the MS method against manual measurements, ultrasonic snow depth sensors, and a weighing gauge using an industry standard method.

### 3.1 Laboratory experiment method and validation

As illustrated in Fig. 3, laboratory experiments were conducted using a DEID disdrometer, a temperature and relative humidity sensor (Vaisala HMP155, sample rate of 1 Hz, accuracy of $\pm 5\%$ relative humidity, temperature of $\pm 1$ °C), a high-precision gravity scale (Sartorius model ENTRIS64-1S with a readability of 0.1 mg and a standard deviation of

0.1 mg), a micropipette (accuracy of 1 % and maximum accuracy of 1.2 µL at the highest volume), a silicon mold, and a freezer with a minimum temperature of $-37$ °C. The precise setup of the DEID is modifiable, but for this study, the thermal camera measured surface temperatures of a hotplate with a dimension of $\approx 9$ cm $\times$ 6 cm at 15 fps with 531 pixels $\times$ 362 pixels, which yields a spatial resolution of about 0.2 mm per pixel on the plate. The thermal camera looked vertically downward at the hotplate at an angle. As a result, the maximum error in area measurement is 1.6 %, and the error in the area was corrected using a custom-made function based on the height and angle of the thermal camera, and details are given in Appendix D. Continuous thermal-camera imagery (recorded at 15 Hz) provided all relevant parameters for calculating hydrometeor mass, except for the effective thermal-conduction coefficient between the hotplate and hydrometeor ($\kappa$), which was determined through laboratory calibration. We used an Infratec thermal camera that writes out infrared binary (IRB) files that store each pixel's absolute temperature. In post-processing, a grey-scale thermal image ranging in intensity from 0 to 255 is created from the IRB files based on a preset infrared temperature range. To determine $\Delta t_{melt}$ and $\Delta T_{melt}$, the temperature range used was $[-40, 0]$ °C, and for the $m$, $A_c$, $\Delta t_{evap}$, and $\Delta T_{evap}$ measurements, the temperature range used was $(0, 85]$ °C, as illustrated in Fig. 3c. In a table-top experiment, the DEID was operated at 85 °C, as determined using the thermal camera. To measure $v_{melt}$ using $h$ and $\Delta t_{melt}$, the DEID was placed in a 0.25 m per side open-topped cubic enclosure within an environment with a near-zero wind-tunnel wind speed of 0.02 m s$^{-1}$, a constant ambient temperature of 18 °C, and a constant relative humidity of 38 %. A total of 80 hemispherically shaped ice particles with a range of known masses and volumes were made in a laboratory freezer using a micropipettor by applying a distilled water droplet to a flat silicon mold. A side view of the thermal image of the hemispherical ice particle sample is provided in Fig. 3b. Droplet volumes of 5, 10, 20, 30, 40, 50, 60, and 70 µL, with 10 samples per volume, were used to manufacture the ice particles. The height of the ice particles $h$ was measured in the freezer as illustrated in Fig. 3d. A U-shaped rigid metallic frame was mounted on a translational stage. A millimeter ruler was attached to one arm of the U frame, and a laser pointer was affixed perpendicularly to the second arm facing towards the ruler on the opposite arm in such a manner that when turned on, the pointer's laser beam would strike a spot on the ruler bar. The pointer was horizontally aligned independently using a precision bubble level. The ice particle was thus positioned between the two arms of the U frame. The ruler scale's purpose was to obtain the vertical height of the ice particles. The U frame was vertically displaced when the ice particle occluded the laser beam and did not reach the ruler scale. Upon emerging at the top of the ice particle, when the beam spot hit the ruler scale, the reading was taken as illustrated in Fig. 3d. Furthermore, ice particle mass was also

**Table 1.** The shape of the sample ice particles made in a laboratory: $\mathcal{V}_{\text{pipette}}$ is the volume of water applied by the pipette, $A_c$ is the cross-sectional area of ice particles measured by the thermal camera on the hotplate, and $R$ is the maximum thickness (radius) of ice particles determined from the laser pointer system shown in Fig. 3.

| $\mathcal{V}_{\text{pipette}}$ (µL) | $A_c$ (mm$^2$) | $R$ (mm) |
|---|---|---|
| 5 | $5.65 \pm 0.09$ | $1.21 \pm 0.07$ |
| 10 | $9.10 \pm 0.09$ | $1.64 \pm 0.03$ |
| 20 | $13.90 \pm 0.11$ | $2.23 \pm 0.08$ |
| 30 | $17.60 \pm 0.09$ | $2.54 \pm 0.03$ |
| 40 | $22.61 \pm 0.11$ | $2.28 \pm 0.07$ |
| 50 | $26.81 \pm 0.17$ | $2.85 \pm 0.05$ |
| 60 | $31.13 \pm 0.16$ | $3.25 \pm 0.09$ |
| 70 | $34.56 \pm 0.14$ | $3.34 \pm 0.08$ |

measured with a gravimetric scale prior to its application on the hotplate. The laboratory ice particle sample dimensions are summarized in Table 1.

Individual frozen droplets were placed on the hotplate, and the cross-sectional area $A_c$, $m$, $\Delta T_{\text{melt}}$, $\Delta T_{\text{evap}}$, $\Delta t_{\text{melt}}$, and $\Delta t_{\text{evap}}$ were measured using the DEID from Eq. (8). $v_{\text{melt}}$ was then calculated using two different formulas $v_{\text{melt}} = h/\Delta t_{\text{melt}}$ and using Eq. (10), where $h$ was determined from the laser pointer system.

### 3.2 Environmental impacts on $v_{\text{melt}}$ measurement: wind-tunnel experiments

For any given ice particle mass $m$, independent of the rate of melting or evaporation, the $\approx m(L_f + L_v)$ constant total quantity of energy is required both to melt and to evaporate a particle from the hotplate. However, the conductive heat rate from the hotplate to the ice particle is a function of environmental conditions through the temperature difference $\Delta T$. To determine how environmental variability affects measurement of the melting velocity $v_{\text{melt}}$, a portable wind tunnel was set on one side of the DEID's hotplate, allowing different wind velocities to pass over the hotplate and ice particles. A pitot-static probe and an automated weather station measured air speed, ambient temperature, and relative humidity. Ice particles of 60 µL (0.06 g) with radius $R = 3.25$ mm and area $A_p = 30.13 \times 10^{-6}$ m$^2$ were placed on the DEID's hotplate. Three experiments were performed. First, the hotplate temperature and the ambient relative humidity were maintained constant at 85 °C and 38 %, respectively, while the wind tunnel was adjusted for air speeds of 0.0, 1.5, 3.0, 4.7, 5.9, and 8.3 m s$^{-1}$. Second, ice particle experiments were performed for surface plate temperatures of 65, 85, and 95 °C for near-zero wind speed and 38 % relative humidity. Finally, the relative humidity was varied to cover 38 %, 68 %, and 91 % with near-zero wind speed and a constant hotplate temperature of 85 °C.

### 3.3 Density of individual frozen saltwater particles

To test the MS method on a wide range of particle densities, a method was required to produce particles of varying densities. This was done by creating frozen droplets by adding sodium chloride in a distilled water solution (see Table 4 for the percentage of sodium chloride). The densities of these particles were then measured using the DEID and the same methods described above for the pure frozen-water tests.

### 3.4 Field validation experiments

Data were obtained from field experiments conducted during the winter between October 2020 and April 2021 comprising 17 snowfall events in the upper Little Cottonwood Canyon, Utah, USA, at the Alta Ski Area's mid-Collins snow-study plot (Alcott and Steenburgh, 2010) (40.5763° N, 111.6383° W; 2920 m above sea level). A 10 m crank-up measurement tower at the site included a DEID (sampling rate of 15 Hz) for measurement of microphysical properties of snowflakes and a 3D sonic anemometer (Campbell Scientific, Inc. CSAT3, sampling rate of 20 Hz and accuracy of $\pm 0.05$ m s$^{-1}$). In addition to the DEID, a particle imaging system, consisting of a laser sheet with a sampling volume of 10 cm $\times$ 18 cm $\times$ 7 cm, was simultaneously deployed and oriented normal to the viewing angle of a Nikon D850 single-lens reflex (SLR) camera as shown in Fig. 4a. The SLR camera recorded 1920 pixel $\times$ 1080 pixel images at a spatial resolution of $\approx 160$ µm per pixel at 120 fps within a vertical laser sheet created using three 10 W 520 nm diode lasers and a collimator lens. The laser beam spread angle of $\approx 6.8°$ allowed for a light sheet with near-constant thickness of $\approx 7$ cm throughout the region of interest. A single-focal-length Nikon AF-S VR Micro-Nikkor 105 mm $f/2.8$ G IF-ED lens permitted a depth of field greater than the thickness of the laser light sheet. The DEID was deployed 2 cm below the lower end of the laser sheet, which permits measurement of the microphysical properties of snowflakes that pass the laser sheet and fall on the hotplate. A Vaisala HMP155 temperature and relative humidity sensor (1 Hz sampling rate) was also located on the tower and maintained at a height of approximately 1.5 m above the new snow level. At approximately the same height, a Campbell Scientific, Inc. CSAT3 3D sonic anemometer was deployed (sampling rate of 20 Hz).

Images from the SLR camera were combined with mass measurements from the DEID to compute snowflake density in the field and validate the MS method. With this SLR–DEID method, the geometrical volume of each free-falling snowflake was estimated using images from the SLR camera. The mass of each hydrometeor was determined by following individual snowflakes through the laser sheet until they hit the DEID hotplate. This method was applied to approximately 1000 snowflakes. Selected thermal images of aggre-

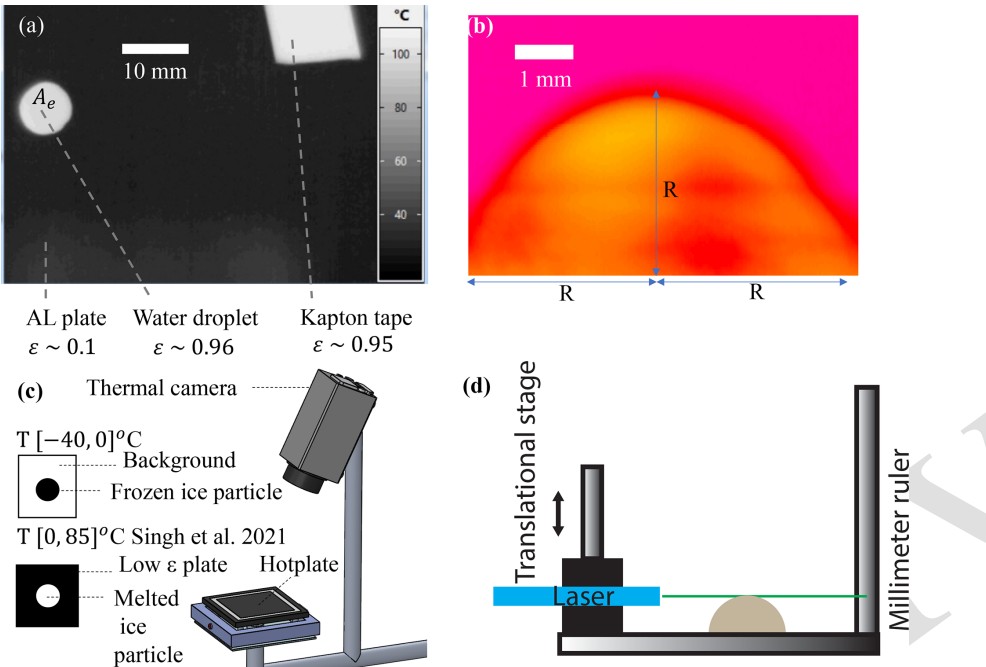

**Figure 3.** Details of the DEID MS measurement technique. **(a)** A hemispherical ice particle applied to the hotplate seen as a bright circular region after melting alongside a rectangular piece of Kapton tape ($\epsilon \approx 0.95$) used to measure the hotplate surface temperature ($T_p$). **(b)** Surface-temperature contour plot of the side view of an ice particle obtained using the thermal camera at $t = 0$ for a temperature range of $[-40,0]\,°C$. **(c)** Schematic of the DEID showing the imaging of melting and evaporating particles, respectively. The black and white contrast of the ice and water particles is optimized by adjusting the camera's temperature range. **(d)** Schematic of the ice particle height measurement apparatus.

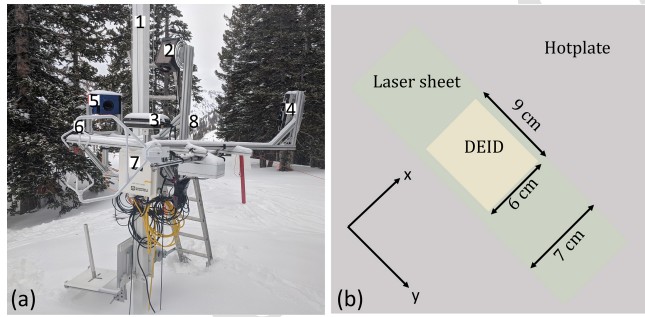

**Figure 4. (a)** Field experiment setup for measurements of microphysical properties of snowflakes and snowflake visualization. The experimental setup consists of (1) a 20 m tower, (2) a thermal camera, (3) a hotplate, (4) three 10 W lasers and an optical lens, (5) a D850 Nikon SLR camera, (6) a 3D sonic anemometer, (7) a data logger and computer, and (8) a relative humidity and temperature sensor. **(b)** Top-view schematic showing the co-location of the laser system and DEID hotplate.

gate snowflakes and graupel on the hotplate are illustrated in Appendix B.

## 3.5 Hydrometeor density calculation exploiting concurrent imagery during their fall

The geometrical volume $\mathcal{V}_{SLR}$ of a snowflake can be estimated independently of the DEID by imaging falling snowflakes using the particle tracking system discussed above. In this case, the density is determined from

$$\rho_{SLR\text{-}DEID} = \frac{m}{\mathcal{V}_{SLR}}, \tag{19}$$

where snowflake mass $m$ is determined with the DEID from Eq. (8).

We categorized five snowflake habits based on the international classification for seasonal snow on the ground (Fierz et al., 2009; Praz et al., 2017): planar crystal (combining stellar and plates), graupel (combining hail and graupel), columnar crystal, aggregate (combining irregular crystal), and small particles. Graupel and small-particle crystals were classified based on size. Each snowflake category contained approximately 200 samples. Taking advantage of how snowflakes rotate while falling, a single camera with multiple images was found to represent a 3D picture of a snowflake and provide geometrical volume more accurately than using a single image (Li et al., 2022). Five sequentially selected images of each crystal type are illustrated in Fig. 5a.

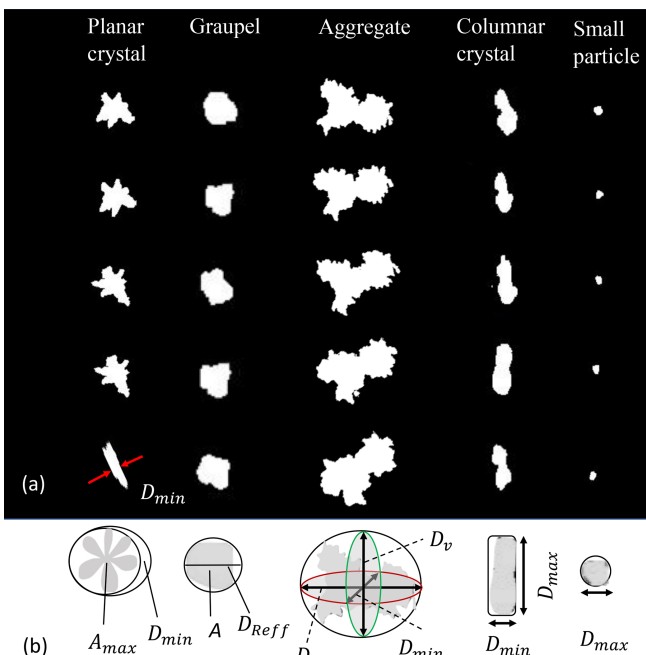

**Figure 5. (a)** For five snowflake types, five images of each snowflake separated due to rotation just prior to settling on the DEID's hotplate. **(b)** Volume measurement method.

A schematic showing how snowflake volume is computed is illustrated in Fig. 5b and formulated in Table 2. The volume of planar crystals was approximated as a disk; hence, $\mathcal{V}_{SLR} = A_{max}D_{min}$. Here, $A_{max}$ is the maximum area of a planar crystal in a 2D plane among all the images that are visible to the camera, and $D_{min}$ is the minimum dimension (representing the thickness of the disk) in the 2D plane as indicated in Fig. 5a, column I, row V. The volume of graupel was estimated as a sphere, $\mathcal{V}_{SLR} = \pi/6 D_{Reff}^3$, where $D_{Reff}$ is the effective circular diameter estimated using the SLR camera imaging the snowflake falling in the air and estimated as $D_{Reff} = \sqrt{\frac{4}{\pi}\overline{A}}$. $\overline{A}$ is the 2D mean area of all images as illustrated in Fig. 5a, column II. The volume of aggregates was estimated by fitting an ellipsoid such that $\mathcal{V}_{SLR} = \frac{\pi}{6}D_{max}D_{min}D_v$, where $D_{max}$, $D_{min}$, and $D_v$ are the lengths of three mutually perpendicular ellipsoid axes as illustrated in Fig. 5b. The volume of columnar crystals (solid cylinder) was estimated from $\mathcal{V}_{SLR} = \frac{\pi}{4}\overline{D}_{min}^2\overline{D}_{max}$, where $\overline{D}_{max}$ and $\overline{D}_{min}$ are the average of the maximum and minimum dimensions of five images as illustrated in Fig. 5a, column III. The volumes of small particles were estimated using a spherical volume $\mathcal{V}_{SLR} = \frac{\pi}{6}\overline{D}_{max}^3$, where $\overline{D}_{max}$ is the average of the maximum dimension of five small-particle images (Fig. 5a, column V).

**Table 2.** Classification of snowflakes and corresponding volume estimation based on crystal geometry.

| Snow crystal type | Geometrical shape | Volume ($\mathcal{V}_{SLR}$) |
|---|---|---|
| Planar | Disk | $A_{max}D_{min}$ |
| Columnar | Solid cylinder | $(\pi/4)\overline{D}_{min}^2\overline{D}_{max}$ |
| Graupel | Spherical | $(\pi/6)D_{eff}^3$ |
| Small particle | Spherical | $(\pi/6)\overline{D}_{max}^3$ |
| Aggregate | Ellipsoid | $(\pi/6)D_{max}D_{min}D_v$ |

## 3.6 Manual measurements of the snowpack: bulk density snowpack calculations

The mean bulk density of a fresh snowpack ($\rho_s^m$) can also be determined using manual measurements of the ratio of the snow water equivalent depth (SWE) to the new snow depth ($H$). This can be done with the DEID by recalling that $SWE = k\Delta m/\rho_w A_{hp}$ and $H = k\Delta m/\rho_s^m A_{hp}$, where

$$\rho_s^m = \rho_w \frac{SWE}{H}. \tag{20}$$

Here, $\rho_w$ is the density of water of $1000\,\mathrm{kg\,m^{-3}}$. At the Alta-Collins snow-study plot these measurements were obtained every 12 h. Note that changes within the snowpack due to processes such as densification, heat transfer, wind shear, etc. are not considered here as our analyses are limited to the consideration of freshly fallen snow.

For further comparison, the average snowpack density can also be estimated at hourly intervals based on measurements obtained from the ETI Instrument Systems Noah II precipitation weighing gauge sensor ($SWE_{ETI}$) and the snow depth from the Campbell Scientific, Inc. SR50 ultrasonic snow depth sensor. The ETI and SR50 sensors were deployed 4 m from the DEID at the Alta-Collins site. A windshield was deployed around the ETI bucket to increase catchment efficiency. The ETI reported SWE measurements once every hour with a resolution, threshold, and accuracy of 0.25, 0.25, and ±0.25 mm, respectively. The SR50 sensor recorded snow depth every hour to provide running totals of snow depth. The measurement range of the ultrasonic sensor was 0.5 to 10 m with an accuracy of 0.4 % and a resolution of 0.1 mm. Raw DEID data sampled at a rate of 15 Hz were integrated to produce hourly measurements for comparison with ETI data. ETI and SR50 data were collected throughout the winter of 2020, but data from 07:00 UTC on 12 December to 19:00 UTC on 12 December 2020 were used to compare SWE and snow accumulation with the DEID obtained using the MS density measurement method.

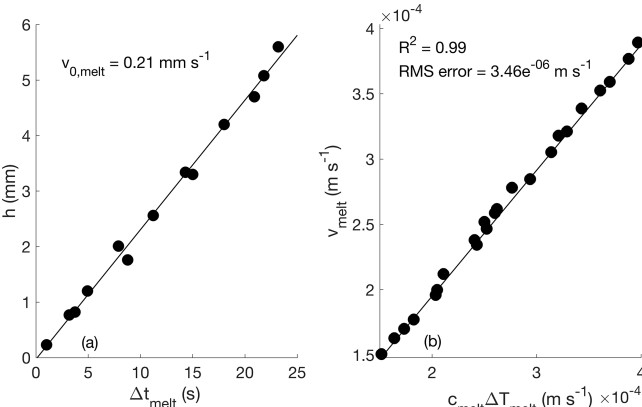

**Figure 6. (a)** Ice particle height as a function of $\Delta t_{melt}$. The slope of this line determines $v_{0,melt}$, the melting speed under fixed standard conditions. That is, ice particles of different maximum thicknesses [0.22, 5.6] mm and their melting time at a plate temperature of 85 °C, near-zero wind velocity (0.05 m s$^{-1}$), and 38 % relative humidity. **(b)** Plot of $v_{melt}$ versus $c_{melt}\Delta T_{melt}$ illustrating the validity of Eq. (10). $v_{melt}$ is determined directly from measurements of particle maximum thickness and melting time, $\Delta t_{melt}$, and then compared to $c_{melt}\Delta T_{melt}$ for a range of environmental conditions.

## 4 Results

### 4.1 MS density method laboratory calibration

In order to use the MS method for determining density, the melting calibration constant, $c_{melt}$, and the calibration constant, $c$, in Eqs. (10) and (12), respectively, must first be determined empirically. To do this, 80 ice particles with different sizes and masses were applied to the DEID hotplate. Furthermore, the experiments were conducted in an environmentally controlled chamber with the air temperature fixed at 18 °C, near-zero wind velocity (0.05 m s$^{-1}$), a hotplate temperature of 85 °C, and 38 % relative humidity. Results from experiments conducted at these standard conditions are identified with subscript 0. The variables $\Delta t_{melt}$, $\Delta t_{evap}$, $\Delta T_{melt}$, and $\Delta T_{evap}$ for each particle were determined using the thermal camera, while $h$ was measured directly using the laser pointer system. The measured values of $h$ and $\Delta t_{melt}$ are plotted in Fig. 6a. The slope of the $h-\Delta t_{melt}$ curve is $v_{melt}$, which is approximately constant ($v_{0,melt} = 2.11 \pm 0.10 \times 10^{-4}$ m s$^{-1}$) because the experiments were performed with ice particles in an environmentally controlled chamber where the average measured value of $\Delta T_{0,melt}$ was also found to be approximately constant ($101.43 \pm 0.82$ K) TS2.

The measured $v_{melt}$ and $\Delta T_{melt}$ for each particle can be substituted into Eq. (10) to solve for $c_{melt}$, and then the constant $c$ can easily be determined from Eq. (13). This was done, and the results were averaged over all 80 samples, yielding $c_{melt} = 2.08 \pm 0.11 \times 10^{-6}$ m$^{-1}$ K$^{-1}$ and $c = 7.14 \pm 0.33 \times 10^4$ TS3. With a derived value of $c$ and particle mass measured with the DEID, the particle density can now

**Table 3.** Mean and standard deviation of the mass of 80 experimentally manufactured ice particles, with the water droplet volume applied by pipette ($\mathcal{V}_{pipette}$) prior to freezing, determined using a gravimetric scale ($m_{gravity}$) and the DEID ($m_{DEID}$). The density of ice particles estimated using the DEID MS method – $\rho_{MS}^{ice}$. The melting velocity under the standard conditions described in the text – $v_{0,melt}$ (m s$^{-1}$).

| $\mathcal{V}_{pipette}$ ($\mu$L) | $m_{gravity}$ (mg) | $m_{DEID}$ (mg) | $\rho_{MS}^{ice}$ (kg m$^{-3}$) | $v_{0,melt} \times 10^{-4}$ (m s$^{-1}$) |
|---|---|---|---|---|
| 5 | $5.6 \pm 0.4$ | $5.1 \pm 0.2$ | $926 \pm 32$ | $2.15 \pm 0.14$ |
| 10 | $12.3 \pm 0.7$ | $14.0 \pm 0.8$ | $916 \pm 58$ | $2.09 \pm 0.12$ |
| 20 | $21.6 \pm 0.8$ | $22.1 \pm 1.0$ | $942 \pm 61$ | $2.11 \pm 0.12$ |
| 30 | $30.8 \pm 2.7$ | $29.8 \pm 2.0$ | $938 \pm 84$ | $2.16 \pm 0.16$ |
| 40 | $42.0 \pm 3.2$ | $43.1 \pm 3.4$ | $906 \pm 58$ | $2.04 \pm 0.09$ |
| 50 | $53.1 \pm 4.1$ | $52.1 \pm 2.4$ | $901 \pm 67$ | $2.08 \pm 0.11$ |
| 60 | $61.8 \pm 3.8$ | $63.1 \pm 3.8$ | $928 \pm 76$ | $2.12 \pm 0.07$ |
| 70 | $74.1 \pm 4.1$ | $76.2 \pm 4.2$ | $899 \pm 86$ | $2.14 \pm 0.04$ |
| | | | $919 \pm 65$ | $2.11 \pm 0.10$ |

be inferred from Eq. (12). This yields an average ice particle density of $\rho_{MS}^{ice} = 919 \pm 65$ kg m$^{-3}$. This is very close to the expected value of ice density at temperatures near 0 °C (i.e., 917 kg m$^{-3}$).

A summary of the following measured parameters for laboratory-created ice particles is presented in Table 3: $v_{melt}$, $\rho_{MS}^{ice}$, $m_{DEID}$, $m_{gravity}$, and $\mathcal{V}_{pipette}$. Here, $\mathcal{V}_{pipette}$ is the volume of a water droplet, not the ice particle volume. Since these experiments were conducted in an enclosure where environmental variability was negligible, the effect of convective cooling on the measurement of $m$, $A_c$, $\Delta t_{melt}$ $\Delta t_{evap}$, and $\Delta T_{evap}$ did not play a role. However, in nature, winds can affect $\Delta t_{melt}$ $\Delta t_{evap}$, $\Delta T_{evap}$, and $v_{melt}$.

To determine how environmental variability affects $v_{melt}$ and to generalize the validation of Eq. (10), 60 $\mu$L (0.06 g) ice particles with thickness $R = 3.25$ mm and area $A_c = 30.13 \times 10^{-6}$ m$^2$ were placed on the DEID's hotplate, and the wind speed was varied from 0.0 to 8.3 m s$^{-1}$, relative humidity varied from 38 % to 91 %, and plate temperature varied from 65 to 95 °C. $v_{melt}$ was computed using direct measurements as $v_{melt} = R/\Delta t_{melt}$. It was also computed using Eq. (10; i.e., $v_{melt} = c_{melt}\Delta T_{melt}$. Results are shown in Fig. 6b. The coefficient of determination $R^2$ between $v_{melt}$ computed using the two methods is 0.99, and the RMS error is $3.46 \times 10^{-6}$ m s$^{-1}$. Using Eq. (12), the average measured density of 80 ice particles with different shapes and sizes, formed by pipetting water droplets onto arbitrary surfaces with different contact angles under the above-listed environmental conditions and plate temperatures, was $928 \pm 56$ kg m$^{-3}$, which shows the MS method works for a wide range of environmental conditions within experimental uncertainty.

**Table 4.** The density of frozen saltwater particles measured using the MS method and determined theoretically based on a saltwater ratio by weight (percentage of salt and water).

| Percent salt (%) | Percent water (%) | $\rho_{MS}$ (kg m$^{-3}$) | $\rho_{theoretical}$ (kg m$^{-3}$) |
|---|---|---|---|
| 1.47 | 98.52 | $969 \pm 37$ | 1008 |
| 2.75 | 97.24 | $988 \pm 26$ | 1015 |
| 5.07 | 94.92 | $1002 \pm 38$ | 1028 |
| 7.16 | 92.83 | $1018 \pm 24$ | 1040 |
| 8.86 | 91.13 | $1028 \pm 31$ | 1050 |

## 4.2 Density of individual frozen saltwater particles

To test the MS method on a wider range of particle densities, frozen saltwater particles with different salt (NaCl) concentrations were applied to the DEID hotplate. The estimated density of the frozen saltwater particles calculated using the MS method agrees with the density expected from the percentage of salt in the solution ($\rho_{theoretical}$) to within 3 %. The results are summarized in Table 4.

## 4.3 Field evaluation of the MS method for snow particles of different types

We collected data at the Alta Ski Area's mid-Collins snow-study plot from 7 November 2020 to 27 April 2021. During this time, a snow accumulation of 12.35 m and an SWE accumulation of 1.38 m were observed with the DEID. The ambient air temperature varied from −21 to 2 °C, relative humidity varied from 64 % to 97 %, and wind speed varied from 0.2 to 8 m s$^{-1}$. Generally, the observed densities of freshly fallen individual snowflakes varied from 9 to 495 kg m$^{-3}$. The average densities of each storm varied from 35 to 115 kg m$^{-3}$. Figure 7 shows estimated snowflake densities using both the SLR–DEID method and the MS method for five types of snowflakes. A comparison between the MS and SLR–DEID density methods for five crystal types is summarized in Table 5. The coefficient of determination between the two methods is the highest for small particles and graupel and the lowest for aggregates. The measured size ($D_{eff}$) of each type of crystal is summarized in Table 5. The uncertainty that arises between the two methods for aggregate snowflakes may be due to the SLR–DEID method used to estimate the geometrical volume because aggregates have a more irregular shape than the other snow crystal types. For all snowflakes, the mean estimated density is $131 \pm 83$ kg m$^{-3}$ using the SLR–DEID method and $142 \pm 87$ kg m$^{-3}$ using the MS method, yielding an uncertainty of 3.9 % and an $R^2$ of 0.95.

## 4.4 Frequency distribution of individual hydrometeor densities

Figure 8a shows a probability distribution function (PDF) for the densities of individual snowflakes using the MS method applied to data acquired at the Alta-Collins snow-study plot for seven snow storms selected to encompass a broad range of environmental conditions: mean ambient temperatures [−13.45, −4.82] °C, mean wind speeds [0.30, 0.89] m s$^{-1}$, and mean relative humidities [72 %, 91 %] as listed in Table 6. The kurtosis (Kr) and skewness (Sk) of the normalized density distribution functions vary from 2.02 to 2.42 and from 0.01 to 0.10, respectively. We found that the PDFs of snow density are symmetric with respect to the mean (Sk = 0.01) and near Gaussian (Kr = 2.41) when the ambient temperature is the lowest, while Sk = 0.10 and Kr = 2.02 when the ambient temperature is the highest, which is shown in Table 6. Figure 8b includes results from snowflake densities computed assuming a spherical-particle volume but also using the DEID, as done in Rees et al. (2021). The spherical assumption underestimates snowflake density by a factor of $\approx 1.5$ compared to the MS method.

## 4.5 Validation of SWE measurements

SWE determined with the DEID can be compared to manual measurements collected at the Alta-Collins study plot. Since manual measurements are made infrequently at intervals of 12 h, the comparisons are made on a storm-by-storm basis as shown in Fig. 9. The relationship between DEID observations and the bulk standard manual measurement techniques is shown in Fig. 9a. A best-fit relationship between the two methods yields an $R^2$ of 0.994 with a slope of $0.94 \pm 0.04$.

The accumulated SWE integrated over 1 min intervals is compared to ETI data in Fig. 9b. DEID SWE accumulation observations match those from the ETI gauge to within $\pm 6$ % over the 12 h measurement period (storm duration). SWE accumulation measured by the DEID is slightly higher than that obtained by the ETI because the minimum resolution of the ETI is 0.254 mm, whereas the minimum DEID resolution is 0.001 mm (Singh et al., 2021). Furthermore, the ETI gauge has been shown to undercatch snowflakes compared to the DEID under high-wind conditions (Singh et al., 2021).

## 4.6 Validation of snow depth measurements

Snow accumulation ($H$) was computed with the DEID using Eq. (18) and $H = \dot{H} \times \Delta t_{res}$, where 1 min average MS density was used. The total snow accumulated in each storm measured using the DEID MS density method compares well with manual measurements obtained every 12 h at the Alta-Collins snow-study plot with an $R^2$ value of 0.983 and a slope of $1.12 \pm 0.07$ as shown in Fig. 10a. A 12 h period with snowfall between 07:00 and 19:00 UTC on 12 December 2020 is shown in Fig. 10b. There is also good agreement

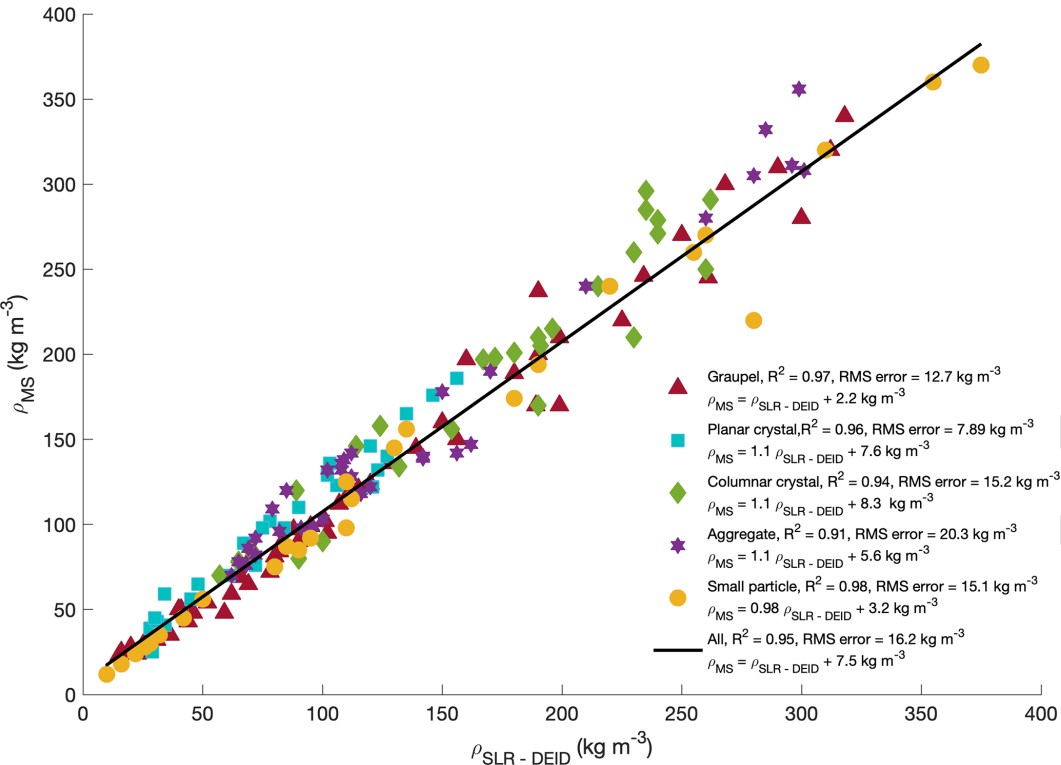

**Figure 7.** Measured density of a range of snowflake types using the particle imaging system compared with values obtained using the DEID MS method with associated coefficients of determination, slopes, and RMSE errors.

**Table 5.** Comparison between two density methods, MS and SLR–DEID, of five types of snow crystals. Range of $D_{\mathrm{eff}}$ for each type of crystal.

| Snow crystal type | $D_{\mathrm{eff}}$ (mm) | $\rho_{\mathrm{MS}}$ (kg m$^{-3}$) | $\rho_{\mathrm{SLR-DEID}}$ (kg m$^{-3}$) | Uncertainty (%) | $R^2$ |
|---|---|---|---|---|---|
| Planar | 2.4–4.3 | $89 \pm 40$ | $98 \pm 46$ | 9.6 | 0.96 |
| Columnar | 1.4–3.4 | $157 \pm 82$ | $140 \pm 92$ | 11.4 | 0.95 |
| Graupel | 1.4–4.6 | $120 \pm 87$ | $130 \pm 89$ | 3.7 | 0.97 |
| Small particle | 0.8–1.2 | $141 \pm 109$ | $138 \pm 110$ | 2.1 | 0.98 |
| Aggregate | 3.1–10.2 | $188 \pm 72$ | $170 \pm 64$ | 10.1 | 0.91 |

to within $\pm 5\,\%$ between snow accumulation measurements obtained from the DEID using the MS density method and those obtained using an ultrasonic snow depth sensor obtained once per hour. The bulk density of a fresh snowpack can differ from the average density of individual snowflakes and the 1 min average because snowflake settling and compaction on the ground depend on considerations such as their settling characteristics, fall angle, wind speed, the structure of snowflakes, and the ambient temperature. We do not account for these processes in the calculation of the volume of freshly fallen snow layers as the impacts are largely unknown. Nonetheless, the bulk density of the snowpack measured during 17 storms using the DEID MS method can also be compared with manual gravimetric snow density measure-

ments of SWE depth (SWE$_m$) and snow depth ($H$) (Eq. 20). The $R^2$ value relating the DEID and manual bulk density measurements is 0.88 with a slope of $0.90 \pm 0.15$, as shown in Fig. 11. The implication of these two comparisons, somewhat surprisingly, is that the DEID reproduces measurements of freshly fallen snowpack density and accumulation, made with more traditional techniques, without considering the quite complex physics of how individual snowflakes pack and stack.

## 5   Conclusions

Automated determination of the density of individual snowflakes has been a long-standing challenge. In this study,

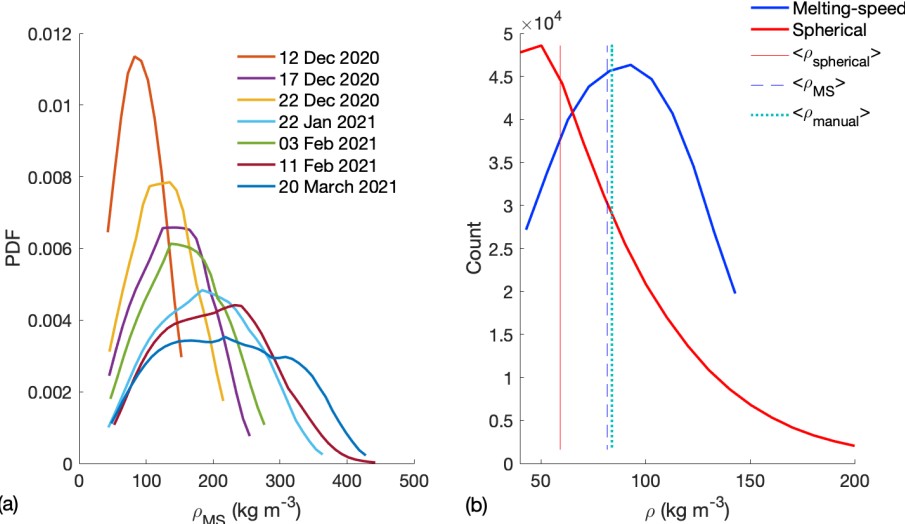

**Figure 8. (a)** Probability distribution functions of the density of individual snowflakes measured using the MS method for seven storms. **(b)** Comparison between the densities determined with the two methods employing DEID measurements alone, the MS method, and the spherical-particle method. The solid line, dashed line, and dotted line are the mean densities of a storm using a spherical method, an MS method and manual measurements, respectively. Note that the MS method and manual measurement are compared without knowing the information about snowpack, as mentioned earlier.

**Table 6.** Summary of DEID-derived snowflake parameters for a series of storms in Alta, Utah. Mean diameter $\overline{D}_{\mathrm{eff}}$, mean density $\overline{\rho}_{\mathrm{MS}}$, and total number of snowflakes $N$ captured during seven storms with a mean ambient temperature $\overline{T}_{\mathrm{amb}}$, mean wind speed $\overline{U}$, mean relative humidity $\overline{RH}$, total snow depth, and total accumulated snow water equivalent for seven storms using the DEID. Kr is the kurtosis and Sk the skewness of the density distribution.

| Storm day and duration (hours) | $N$ | $\overline{D}_{\mathrm{eff}}$ (mm) | $\overline{\rho}_{\mathrm{MS}}$ (kg m$^{-3}$) | $\overline{T}_{\mathrm{amb}}$ (°C) | $\overline{U}$ (m s$^{-1}$) | $\overline{RH}$ (%) | Total snow (mm) | Total SWE (mm) | Kr | Sk |
|---|---|---|---|---|---|---|---|---|---|---|
| 12 December 2020; 20 | 242 643 | $1.60 \pm 0.74$ | 41.9 | $-13.5 \pm 1.4$ | 0.58 | 85.2 | 292 | 10.66 | 2.42 | 0.01 |
| 17 December 2020; 25 | 482 152 | $1.50 \pm 0.71$ | 56.3 | $-6.8 \pm 2.3$ | 0.30 | 90.9 | 413 | 27.00 | 2.18 | −0.07 |
| 22 December 2020; 24 | 251 532 | $1.50 \pm 0.71$ | 49.2 | $-12.1 \pm 4.7$ | 0.76 | 81.2 | 314 | 15.38 | 2.39 | 0.03 |
| 22 January 2021; 45 | 570 590 | $1.50 \pm 0.68$ | 90.5 | $-6.8 \pm 2.3$ | 0.65 | 90.3 | 457 | 34.37 | 2.14 | 0.05 |
| 3 February 2021; 62 | 653 976 | $1.40 \pm 0.65$ | 65.3 | $-10.0 \pm 2.5$ | 0.76 | 89.6 | 488 | 31.75 | 2.27 | 0.02 |
| 11 February 2021; 67 | 1 275 102 | $1.50 \pm 0.59$ | 56.3 | $-7.3 \pm 3.2$ | 0.56 | 86.6 | 862 | 51.81 | 2.16 | 0.03 |
| 20 March 2021; 24 | 629 870 | $1.60 \pm 0.78$ | 88.8 | $-4.8 \pm 3.1$ | 0.89 | 71.7 | 425 | 35.35 | 2.02 | 0.10 |

we present a novel method for accomplishing this goal that exploits a new hotplate instrument, the Differential Emissivity Imaging Disdrometer or DEID, which has previously been shown to be capable of obtaining highly accurate direct measurements of particle mass. Particle-by-particle density estimates are obtained from measurements of particle mass, particle contact area onto the hotplate, and an estimate of the particle's effective thickness using melting speed and melting time. A particle's effective thickness normal to the hotplate is a product of the concept of a melting speed and melting time from which, in combination with a particle's contact area, an estimate of particle volume and hence its density can be obtained. For individual hydrometeors, this melting speed method was validated using laboratory ice particles of known

density showing a maximum uncertainty of 6.3 %, as well as with videos from field measurements of a range of naturally falling snowflakes with an uncertainty of 3.7 %. DEID observations at a high-elevation mountain site of snow water equivalent (SWE) accumulation, snow depth accumulation ($H$), and bulk snow density ($\overline{\rho}_{\mathrm{MS}}$) from 17 storms taken at the Alta-Collins snow-study plot at the Alta Ski Area during the winter of 2020–2021 agreed well with traditional manual technique measurements with $R^2$ values of 0.994, 0.983, and 0.88 and slopes of $0.94 \pm 0.04$, $1.12 \pm 0.07$, and $0.90 \pm 0.15$, respectively, independent of environmental conditions, including wind speed, ambient temperature, relative humidity, and hotplate temperature. We acknowledge that these bulk snow results are surprisingly good given the com-

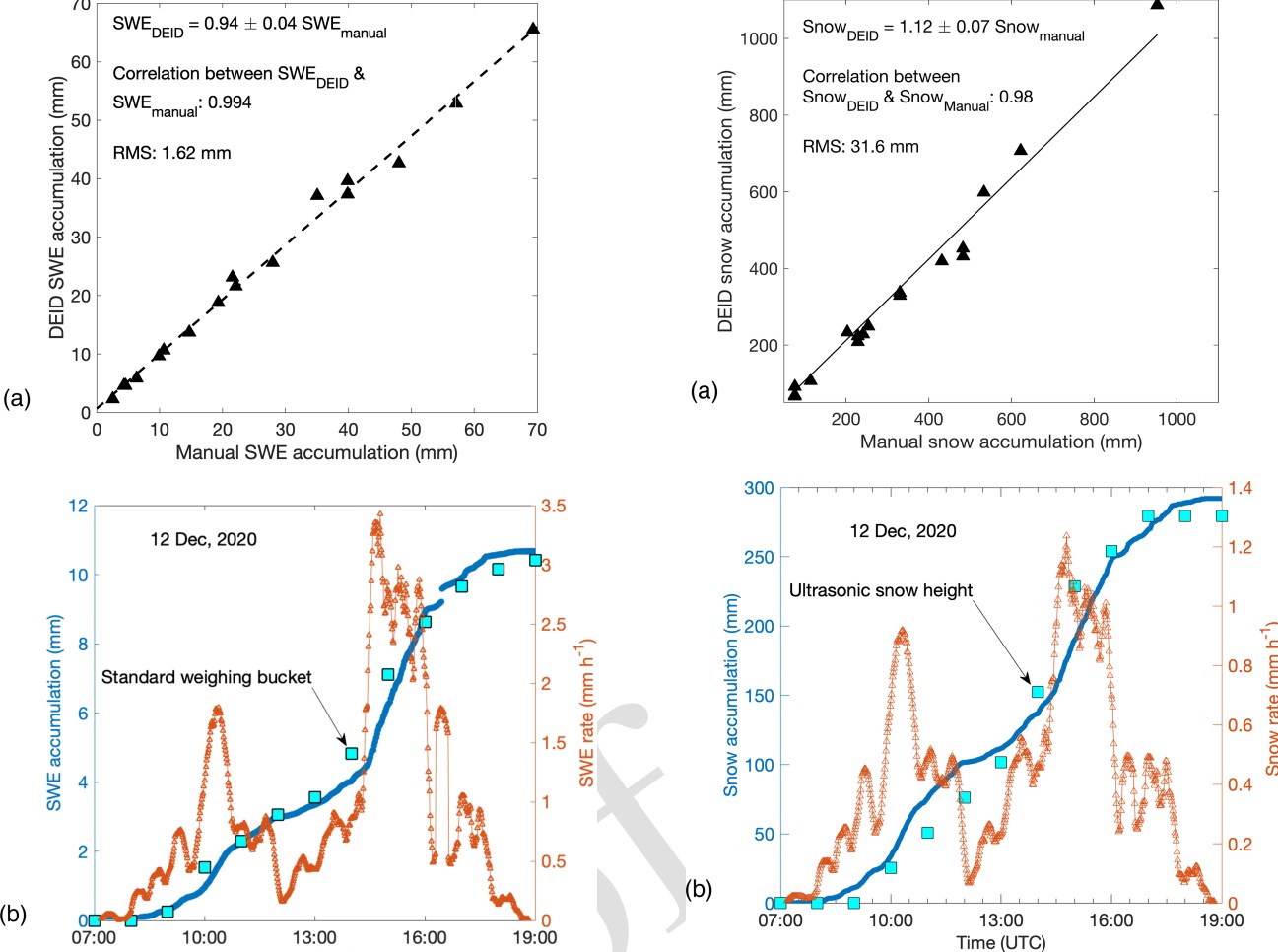

**Figure 9. (a)** SWE accumulation from DEID and manual measurements for 17 storms. Each data point represents an individual storm. Manual measurements were made every 12 h (11:00 and 23:00 UTC), and the DEID was sampled at 15 Hz. **(b)** Time series of SWE accumulation and SWE rate measured using the DEID and ETI gauge. Each DEID data point represents a 1 min average and each ETI gauge data point a 1 h average.

plexity of the processes associated with snow once it hits the ground. We speculate that the snow depth accumulation and bulk snow density results imply minimal snowflake overlap in the snowpack. That is, the sum of the heights of the snowflakes is equal to the total depth. It also implies that the fresh snowpack minimally settles down during the short duration (12 h) associated with the manual measurements. In reality, there is overlap, but on average, the snowflakes are approximately non-overlapping. Our results imply that the density of the snowpack is equal to the average density of individual snowflakes, which means that the snowflakes "pack" with a density similar to the density of the snowflake. We note that sometimes this is not true, for example, when we have an error of 15 %.

**Figure 10. (a)** For 17 storms, comparison between snow accumulation from the DEID obtained using the MS density method and from manual measurements made every 12 h (11:00 and 23:00 UTC). Each data point represents a storm. **(b)** Time series of snow accumulation and snowfall rate measured using 1 min averaged DEID results and 1 h averaged ultrasonic snow depth sensor measurements during a storm on 12 December 2020.

Using the melting speed method, the utility of the DEID design can be extended from hydrometeor mass measurement to measurement of the density of irregularly shaped hydrometeors, in real time with high accuracy. Such information, whether taken on a particle-by-particle basis or assessed as a cumulative bulk quantity, can be applied to high-resolution measurement of vertical density variability in the snowpack (e.g., Morrison et al., 2023) – critical for the assessment of its stability – and to studies of snow metamorphism, the assessment of transitions between rain and snow, determinations of rates of hydrometeor settling and its response to turbulence, and the scattering of visible light and radar pulses by snow particles.

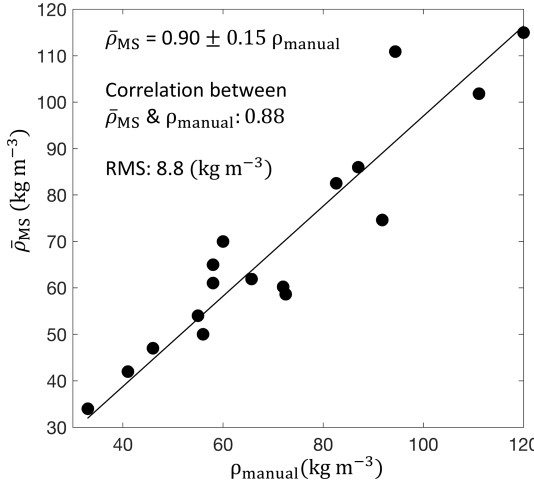

**Figure 11.** For 17 storms, comparison between bulk density of snowpack from the DEID obtained using the MS density method and from manual gravimetric snow density measurements made every 12 h (11:00 and 23:00 UTC). Each data point represents a storm.

## Appendix A: Latent heat of vaporization calculation

The latent heat of vaporization of water depends on temperature and may be written as $L_v(T) = (2.501 - 0.00237 \times T) \times 10^6 \, \text{J kg}^{-1}$, where $T$ is in degrees Celsius (Stull, 2012). In our case, water droplets evaporate at temperatures from 0 to 85 °C. A sample time series of the temperature of a water droplet during evaporation is plotted in Fig. A1. The estimated $L_v$ based on the time series of temperature of water droplets is $2.32 \pm 0.02 \times 10^6 \, \text{J kg}^{-1}$.

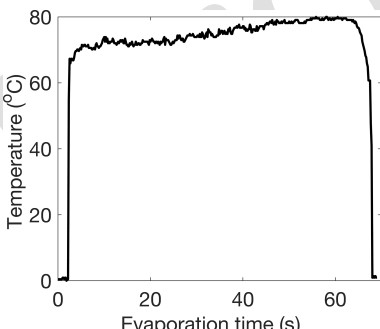

**Figure A1.** Sample time series of the temperature of the melted portion of an ice particle after being placed on the hotplate. A 60 μL water droplet volume was used to make the ice particle shown here.

## Appendix B: Relation between the melting and evaporation statistics

Using Eq. (7), the mass associated with the $ij$th pixel can be estimated separately during melting and evaporation as shown in Eqs. (B1) and (B2) below, respectively.

$$m_{ij} = \frac{\kappa}{C_{ice}T_0 + L_f} \int_0^{\Delta t_{ij,\text{melt}}} \Delta T_{ij,\text{melt}}(t) A_{ij}(t) \, dt \quad \text{(B1)}$$

$$m_{ij} = \frac{\kappa}{C_{ij,\text{liq}}T_p + L_v} \int_0^{\Delta t_{ij,\text{evap}}} \Delta T_{ij,\text{evap}}(t) A_{ij}(t) \, dt \quad \text{(B2)}$$

Here, $A_{ij}$ is the contact area of the single pixel, which is constant. By using Eqs. (B1) and (B2), we may write

$$\frac{\Delta T_{ij,\text{melt}} \Delta t_{ij,\text{melt}}}{L_{ff}} = \frac{\Delta T_{ij,\text{evap}} \Delta t_{ij,\text{evap}}}{L_{vv}}. \quad \text{(B3)}$$

In Eq. (B3), $L_{ff} = C_{ice}T_0 + L_f$ and $L_{vv} = C_{liq}T_p + L_v$.

Equation (B3) can be re-written for all pixels associated with hydrometers using double summation.

$$\sum_i \sum_j \frac{\Delta T_{ij,\text{melt}} \Delta t_{ij,\text{melt}}}{L_{ff}}$$
$$= \sum_i \sum_j \frac{\Delta T_{ij,\text{evap}} \Delta t_{ij,\text{evap}}}{L_{vv}} \quad \text{(B4)}$$

Based on experimental tests, we assume $\Delta t_{ij,\text{melt}} \approx \Delta t_{\text{melt}}$ and $\Delta t_{ij,\text{evap}} \approx \Delta t_{\text{evap}}$ (Eq. B4), which yields

$$\frac{\Delta T_{\text{melt}} \Delta t_{\text{melt}}}{L_{ff}} \approx \frac{\Delta T_{\text{evap}} \Delta t_{\text{evap}}}{L_{vv}}. \quad \text{(B5)}$$

Experimentally determined values for $\Delta T_{\text{melt}}$, $\Delta t_{\text{melt}}$, $\Delta T_{\text{evap}}$, and $\Delta t_{\text{evap}}$ for different ice particle masses and sizes for a large range of environmental conditions are shown in Fig. B1. As expected, the slope between the two terms agrees to within 6.5 % of the ratio of $L_{vv}$ to $L_{ff}$. With the DEID, $\Delta t_{\text{melt}}$ and $\Delta t_{\text{evap}}$ are directly measured using a thermal camera and counting the number of frames between the first frame when an ice particle hits the hotplate and last frame when the particle has completely melted or evaporated.

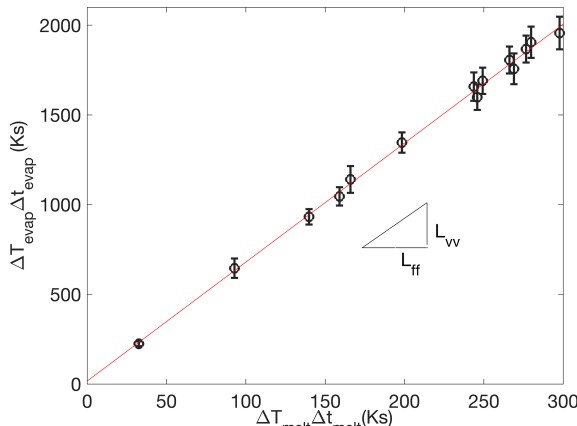

**Figure B1.** Experimentally measured $\Delta t_{\text{melt}}$, $\Delta T_{\text{melt}}$, $\Delta t_{\text{evap}}$, and $\Delta T_{\text{evap}}$ for laboratory ice particles. The slope of the line is quite close to $L_{\text{vv}}/L_{\text{ff}}$. An error bar is associated with 10 samples in each ice particle size.

## Appendix C:  DEID thermal imagery

In this section, we present example thermal images of different types of snowflakes after melting and during evaporation on the DEID hotplate. Figures C1 and C2 show aggregate and graupel snow particles, respectively. These data were taken at the Alta-Collins snow-study plot on 22 December 2020.

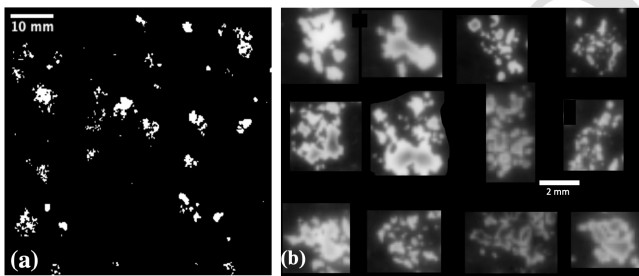

**Figure C1.** **(a)** Black and white binary thermal images of an aggregate type of snowflake at different stages of melting and evaporation on the DEID hotplate observed in Alta, UT. **(b)** Cropped aggregate snowflake images on the hotplate just after melting.

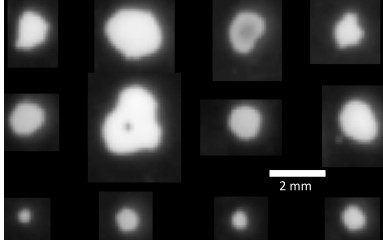

**Figure C2.** Black and white binary thermal images of a graupel type of snowflake during different stages of melting and evaporation on the DEID heated plate observed at Alta.

## Appendix D:  Systematic and random error analysis

Direct measurements made by the DEID include snowflake area, temperature, and evaporation time, for which the respective uncertainties are 1.4 %, 0.3 %, and 1.6 %. Due to the inclination of the thermal camera, the maximum error in the measurement of area is 1.6 %, and it is corrected using a custom-made MATLAB function based on the height and angle of the thermal camera. Both $x$ and $y$ direction pixels were corrected using algorithm $h_{\text{cam}} \tan(\theta_y + j\mathrm{d}\theta)$. Here, $h_{\text{cam}}$ is the height of the camera and $\theta_y = 90 - \theta_H - \theta_V/2$. The angles $\theta_H$ and $\theta_V$ are the thermal camera's horizontal and vertical angles, $\mathrm{d}\theta = \theta_V/j$, where $j$ is the number of pixels in the $y$ direction. A similar method was used for the $x$ direction. The uncertainties in derived quantities (using a standard propagation of uncertainty analysis), such as mass ($m$), height ($h$), and density ($\rho_s$), are 4.3 %, 2.9 %, and 8.6 %, respectively.

## Appendix E:  Separate quantification of area and temperature of ice and liquid

During the post-processing of the data, the temperature ranges of the thermal images were set such that only one phase, either ice or liquid, could be seen on the hotplate. Figure E1a shows a grey thermal image where the temperature of some portions is less than or equal to 0, and some are greater than 0. When using the temperature range $[-40, 0]$ °C, only the ice portion is visible, as seen in Fig. E1b. The sum of all visible areas is $A_{\text{ice}}(t)$, and the spatial mean temperature over all those pixels is called $T_{\text{ice}}(t)$. Similarly, when the temperature range $(0, 85]$ °C is used, only the liquid portion is visible, as seen in Fig. E1c. The sum of all visible areas is $A_{\text{liq}}(t)$, and the spatial mean temperature over all those pixels is called $T_{\text{liq}}(t)$.

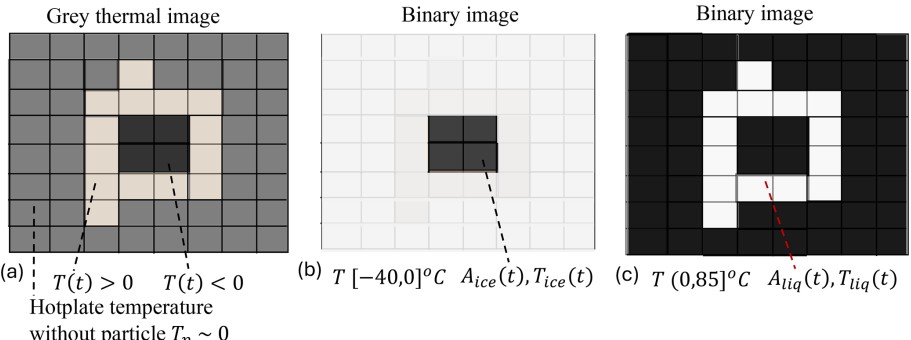

**Figure E1. (a)** Schematic illustrating the measurement area (contact area) and temperature of ice and liquid separately. **(a)** Thermal image of the frozen hydrometeor on the hotplate. The temperatures less than or greater than 0 show the frozen hydrometeor's unmelted and melted portions, respectively. The temperature recorded by the thermal camera for the hotplate is around 0 due to the low emissivity of the plate, where the actual temperature is 85 °C. **(b)** During the post-processing of the data, the temperature range was set to [−40,0] °C, which allows one to see only the area of the unmelted portion (temperatures less than or equal to 0) of the frozen hydrometeor, and the area of the melted portion (temperatures greater than 0) is 0. $T_{ice}(t)$ is the mean temperature of all pixels with a temperature less than or equal to 0, and those pixels contrast well with the background, which allows them to be counted easily. **(c)** During the post-processing of the data, the temperature range was set to (0, 85] °C, allowing us to see only the area of the melted portion (temperature greater than 0) of the hydrometeor. $T_{liq}(t)$ is the mean temperature of all pixels with a temperature greater than 0, and those pixels contrast well with the background, which allows them to be counted easily.

## Appendix F: Image of water droplets on molds

This section presents example images of water droplets on different molds while making laboratory ice particles. Figure F1a and b show water droplets on a flat silicon mold and random surface, respectively. The contact angle between water and the silicon mold is about 90°, and the water droplet looks near hemispherical.

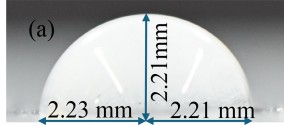 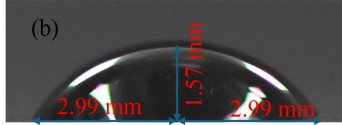

**Figure F1. (a)** A photo of a water droplet on the silicon mold, taken from the side with the SLR camera. **(b)** An image of a water droplet on the random surface. Note the difference in contact angle.

*Code availability.* The data processing codes are protected through a patent and are not available for distribution. The codes used for processing follow the methodologies and equations described herein.

*Data availability.* Data from the ETI Noah II precipitation gauge and ultrasonic snow depth sensor are openly accessible from https://mesowest.utah.edu (University of Utah, 2021). All other datasets are available upon request.

*Author contributions.* DKS, ERP, and TJG designed the experiments, and DKS performed the experiments in consultation with ERP and TJG. All authors analyzed the data and contributed to writing and editing the paper.

*Competing interests.* The Differential Emissivity Imaging Disdrometer or DEID is used and evaluated in this study. The technology is protected through patent US20210172855A1, co-authored by Dhiraj K. Singh, Eric R. Pardyjak, and Timothy J. Garrett, and is commercially available through Particle Flux Analytics, Inc. Timothy J. Garrett is a co-owner of Particle Flux Analytics, Inc., which has a license from the University of Utah to commercialize the DEID.

ther geographical representation in this paper. While Copernicus Publications makes ev-

ery effort to include appropriate place names, the final responsibility lies with the authors.

*Acknowledgements.* We thank Allan Reaburn and his colleagues at Particle Flux Analytics, Inc. for their contributions to the development of the DEID; Dave Richards, Jonathan Morgan, and the Alta Ski Patrol for field support; and Spencer Donovan and Travis Morrison for their contributions to the experimental setup in the field.

*Financial support.* This research has been supported by the Directorate for Geosciences (grant nos. PDM-1841870 and PDM-2210179).

*Review statement.* This paper was edited by Stefan Kneifel and reviewed by Thomas Kuhn and one anonymous referee.

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

**Remarks from the typesetter**

TS1    Due to the requested changes, we have to forward your requests to the handling editor for approval. To explain the corrections needed to the editor, please send me the reason why these corrections are necessary. Please note that the status of your paper will be changed to "Post-review adjustments" until the editor has made their decision. We will keep you informed via email.

TS2    Please see previous remark regarding editor approval.

TS3    Please see previous remark regarding editor approval.

TS4    Please provide date of last access.

TS5    Please provide exact date.

TS6    Please confirm.