# Peer review of "Time-resolved measurements of the densities of individual frozen hydrometeors and of fresh snowfall"

_Atmospheric Measurement Techniques, 2023_

## Referee Comment (RC2)

The manuscript describes measurements with the DEID, the Differential Emissivity Imaging Disdrometer. The instrument has been presented by Singh et al. 2021 who explain that it can "measure the mass, size, density and type of individual hydrometeors as well as their bulk properties." Laboratory and field measurements have been described in that paper, which are very similar to the measurements presented in this manuscript. The authors also cite another study, Rees et al. 2021, which describes mass and density measurements of individual hydrometeors with the DEID in comparison to refined optical imagery data, again like what is presented in this manuscript.

The only new thing with the method is about density estimation. The determination of an effective thickness h allows a different estimation of the volume compared to the estimate based on area equivalent diameter used in Singh et al. 2021 and Rees et al. 2021 (spherical-particle method). Measurement of h is done during a calibration with certain ice particles under certain conditions determining P_0. This P_0 is then used for h, and thus density estimation.

The field campaign provides new data (compared to Singh et al. 2021 and Rees et al. 2021) and comparison of densities in the field. In this campaign a different optical imaging system is used to provide complimentary data on the frozen hydrometeors. This imaging system uses a laser sheet that illuminates the falling hydrometeors from the side, which are then imaged by an SLR camera. Its optical resolution (judged from the images provided) seems similar or only slightly better than the optical resolution of the thermal camera of the DEID.

METHOD to determine mass m (as in Singh et al. 2021 and Rees et al. 2021):
The melted hydrometeor's temperature and area are observed during evaporation, in the laboratory also during the initial faster melting.
From these observations, mass is determined using an equation derived from a heat transfer balance assuming that the heat transferred into the melted hydrometeor equals the sum of the heats required to a) increase the temperature of the hydrometeor and b) melt and c) evaporate the hydrometeor.
0) Determine $\kappa$ (laboratory calibration).
1) Observe during evaporation: $A(t)$, $T\_w(t)$. ($T\_p(t)$ about const)
2) Determine $T\_{p,max}$.
3) Use Eq. (2) to determine m.

METHOD to determine density rho_MS (new in this manuscript):
The density is determined from m divided by $V\_s$, the "volume of a snowflake".
The volume $V\_s$ is estimated as $Ap*h$, where $Ap$ is the "initial snowflake projected area" and h is the "particle's effective thickness in the direction normal to the hotplate". This effective thickness h is estimated as the product of a melting speed and the melting duration. The implied definition (or method of determination) of this melting speed $v\_{melt}$ is h divided by melting duration. $v\_{melt}$ can be determined from $\Delta T\_{melt}$ (average temperature difference between hotplate and frozen part of melting particle) as the two variables are proportional. The proportionality factor includes, besides $\kappa$, the laboratory calibrated coefficient P_0.

For DEID measurements in the field it is easier to observe evaporation duration (instead of the much shorter melting duration), so $\Delta T\_melt$ and melting duration can be replaced by $\Delta T\_evap$ and evaporation duration. Thus, the volume, and with that the density, is estimated from Ap, $\Delta T\_evap$, and evaporation duration (Eq. 8).
0) Determine P_0 (laboratory calibration)
1) Observe during evaporation: T_w(t). (T_p(t) about const)
2) Determine $\Delta T\_evap$
3) Use Eq. (8) to determine density rho_MS

The manuscript continues with a method to determine the average density and derive from that the snow water equivalent precipitation rate and snow accumulation rate.

These methods are explained with a series of equations that are derived step by step. All these equations, however, suffer from unclear, implicit, badly motivated (or not at all), or wrong assumptions. This makes it extremely difficult to check the validity of the methods and to understand under which conditions and for which type of hydrometeors they can be applied. None of these assumptions is adequately discussed. That means that uncertainties related to these assumptions cannot be estimated.

Despite these many issues and unclear assumptions, the results are seemingly good. This is both surprising and interesting and needs better and further discussion. In particular it is interesting how DEID estimated well snowpack densities agree with manually measured density despite the fact that average hydrometeor density is wrongly consider to be the snow pack density.

Due to all the issues with the method and validation measurements using a limited variety of very simply-shaped hydrometeors, most of the conclusions should be re-evaluated after addressing these issues.

Before I can accept this manuscript for publication, I would like to see major changes taking into consideration my feedback.

Below follows a detailed description of some issues, including major issues and some others, that I have seen. I have refrained so far from commenting on other smaller issues with for example language and on commenting in detail all the sections.

**Issues with Equations**
=====================

Eq. (1) --- Issues and questions
===========================
The term representing the heat required to increase the temperature is m*Cw*T_p,max. The specific heat of water Cw is different for ice and for liquid water. This fact seems completely neglected. No value for Cw is given.

Further, I would expect a temperature difference in this term, the amount by which the hydrometeor's temperature is increased. Instead there is T_p,max, which is explained as "maximum surface temperature of the hotplate". It is unclear what T_p,max is and why it is used in this term.
Assuming that T_p,max is the temperature difference by which the hydrometeor's temperature is increased, then this implies or assumes that the whole hydrometeor (m) actually increases by this amount. Is this assumption ok, i.e. does most of the evaporation (and consequential decrease in. m) happen when the melted hydrometeor reaches its highest temperature?

The temperature T_w(t) is the "temperature of the water droplet at time t", whereas area A(t) is the "area of each snowflake at time t". Why does one refer to a droplet and the other to a snowflake?

How is "area" A defined?

The equation includes melting and evaporation. This is clear only from Leqv = Lv + Lf being the sum of the latent heats of fusion and vaporization. The integral in the equation is from time t=0 to Δ t_evap, "the time required to evaporate the water droplet". It is unclear if melting is neglected in this integral or if melting starts at t=0, i.e. if the integral includes melting and evaporation. Does evaporation also start at t=0 or does it start when melting is finished (at t = Δ t_melt)?

κ is determined by evaporating droplets of known mass (in Singh et al. 2021, water droplets of 20µL). The "empirical calibration coefficient" κ seems to represent the specific heat capacity of the contact area between hotplate and melted hydrometeor. κ is (k/d)_eff in Singh et al. 2021, who call this the calibrated value of the term k_w/d_w in their equation for R2 (in text; for R2 see their Fig 2 and Eq 15) without mentioning anywhere what k_w and d_w are.
κ is determined only for liquid droplets and only for one size. Is κ a constant that can be applied to frozen and liquid hydrometeors as done in Eq. (1)?
Melting neglected in Eq. (1)??
Initially, at deposition at t=0, the contact is at a few points only. Most likely it takes a very short time for a liquid layer to form at the hotplate, which provides the contact area needed to apply Eq. (1).

Heat lost from the hydrometeor or melted hydrometeor is neglected. This is not mentioned, motivated or discussed.

Eq. (2) --- Issues and questions
==========================

The time varying temperature difference from Eq. (1) is replaced with the constant Δ T_evap referring to Rees et al. 2021, who should be demonstrating that this can be done. I could not see that Rees et al. 2021 motivate this in any way, they simply do the same substitution from one equation to the next. As with Eq. (1), it is unclear if the integral starting at t=0 includes melting or only shows evaporation. In the former case, Fig2b shows that the temperature difference

cannot be approximated as constant. In the latter case, the integral is over the time range in white in Fig2b and the temperature difference $\Delta T\_evap$ may be approximated as constant.

Eq. (3) --- Issues and questions
==========================

The "particle's effective thickness in the direction normal to the hotplate h" seems to be defined as the height of the quasi hemispherical "droplet" on the hotplate (Fig1). Why is this called "effective" thickness? The melting speed relates h and the time required to "melt an individual snowflake". How is h defined for a "frozen hydrometeor"?

Eq. (3) implies that the "volume of a snowflake" is Ap * h.
When P_0 is later determined (Sects 3.1/3.2 and Sect 4.1), frozen quasi-hemispherical solid ice pellets are used. As an example, one size is generated from water droplets of 60µL volume and stated to have a measured height h of 1.95mm and area A (why is this not called not Ap?) of 0.502 cm2 (L278-279). The volume of these ice pellets is thus (according to Ap * h) 98µL. This cannot be explained by the lower ice density and shows that Ap * h is not the volume of these ice particles. However, this volume is what Eq. (3) is based on. This then affects the following equations too, in particular Eqs (7) and (8), which are used to determine the density of a frozen hydrometeor.

Eq (4) --- Issues and questions
==========================

L_ff should be L_f?

The equation refers only to melting. It seems to implicitly approximate the temperature difference between hotplate and frozen part of hydrometeor as constant $\Delta T\_melt$. Is this justified? I am not sure when looking at Fig2b (the grey part).

What happens to the integral? Why have an integral over A over a constant 1 and not write out what that is? How you get to that integral is also unclear. There seem to be implicit assumption that are not mentioned.

Eq (5) --- Issues and questions
==========================

The concept of "the 'effective mechanical work rate' or mechanical power … required to melt a snowflake" is not adequately introduced, in particular the "constant 'melting pressure'" P_0. Apparently P_0 was found to be constant empirically, and these equations (Eq.5) represent the attempt to derive that with some integrals that seem mathematically dubious. It is unclear what integrals over V_s or A*h between t=0 and $\Delta t\_melt$ as it is mathematically wrong. As with Eq. (4), there seem to be implicit assumptions not mentioned, and the unresolved integral over A appears.

Eq (6)-(9) --- Issues and questions
============================

These equations follow from the above equations, i.e. they are based on the assumptions and approximations of those so that it is difficult to see under which conditions they apply and/or how the density rho_MS is implicitly defined. See also the issues regarding Eq (3) about interpretation of volume and definition of h.

One assumption used to derive Eq. (8) from Eq (7) is the substation shown in L 108 to replace $\Delta$T_melt and $\Delta$t_melt with $\Delta$T_evap and $\Delta$t_evap.
This is based on Eq (4)/Eq (A1) and Eq (A2) with all their assumptions and issues.

Eq (10)-(13) --- Issues and questions
===============================

The accumulated snow water equivalent SWE is simply the snow water equivalent rate of precipitation (from Eq. 10) times $\Delta$t_res. Why use an integral (in Eq 11) if all that is involved is a single value of $\Delta$m and one sampling interval $\Delta$t_res?
Eq. (12) defines an average hydrometeor density of an ensemble of hydrometeors as the sum of their masses divided by the sum of their estimated volumes. I do not agree with calling this "average density of a freshly fallen snowpack layer". If it was the average snowpack density, then Eq. (13) describes the snow accumulation rate, as claimed. But given the definition of this average density, H from Eq. (13) is the height that one would get if all hydrometeors pack exactly (sum of volumes of hydrometeors divided by the surface over which they were collected), i.e. without effectively neither leaving any space in between them nor overlapping. In other words, I don't agree that one can generally determine the height of the snowpack and the accumulation rate from the volumes of the individual hydrometeors without any knowledge of how efficiently they pack.

**DEID laboratory set-up**
========================

The DEID is described briefly here, the previous publication about the DEID and its method (Singh et al. 2021) should contain the details.
The set-up, however, is flexible and in this manuscript is described as a hotplate of 9 cm x 6 cm that is thermally imaged using 531 pixels x 362 pixels of the thermal camera yielding a "spatial resolution" of 0.2 mm/pixel. It is, without mentioning this, assumed that the thermal imagery is looking vertically down at the hotplate. What is not mentioned here, neither for the set-up in Singh et al. 2021, is the actual inclination of around 30 degrees (see Fig 3c). Does this, and if so to what extent, affect the determination of Ap and A?
Fig 3b shows a "Side-view of a surface temperature contour plot of an ice particle obtained using the thermal camera." Has this been obtained with the configuration shown in Fig 3c?

**Sample preparation**
=================

L 164-166 describe the sample preparation to manufacture ice particles for laboratory validation. In a freezer, one of eight different known volumes is applied, using a micropipette, to "a flat silicone mold". Is this a flat silicone sheet rather than a mould suggesting a specific shape? The formed ice particles are referred to as "hemispherical". I guess they will be close to, but not exactly hemispherical, as can be seen, for example, from the value of Ap of 0.502 cm2 (L278-279) for ice particles produced from 60μL of liquid water (a hemisphere of liquid water would have 0.29 cm2). The shape of these sample ice particles should be better described. See also comments on volume and h regarding Eq. (3).

Sect 3.2/Sect 4.1: what is the ambient temperature in the lab when testing environmental variability? In field evaluations (Sect 4.3) the ambient temperature varies. Do results depend on ambient temperature, which for calibrations was plus 18 degree C?

L 274: "Here, Vpipette is volume of the ice particle created by pipetting a known volume of water." Should Vpipette not be the volume of liquid water dispensed from the pipette to generate an ice particle?

Sect. 4.1 last sentence (L 283-285): The "80 ice particles with different shapes and sizes" were presumably all of the same size and same shape. I am assuming this as it is in the same paragraph as the description of test particles as produced from 60μL of liquid water. Else, what does "different shapes and sizes" refer to?

Tab 3: What is "Percentage of salt"?

**Uncertainties**
===========

There is no discussion of how measurement uncertainties affect the measurements of mass and density.

Errors in A as determined by thermal imagery?

Micropipette accuracy?  L 152: "1.00/1.20 %/μL", what does that mean? 1% or 1% for each 1.2 μL; 1% or 1.2 μL (when 1% and when 1.2 μL)?

Accuracy of measurements of h?

Accuracy of temperature measurements?

Maybe less important: accuracy of time measurements? With 15 fps I guess error in time is on the order of 1/15 s.

**Sect 3.5 --- geometrical volume estimates**
=====================================

The explanation of how volume is determined is not sufficient. The quantities Dmin, Dmax, A, mean(A), and Dv are not defined adequately. Planar crystals are approximated as a disc. Is this disc a cylinder (as suggested by Fig 5b) or a column with base Amax (as suggested by the definition of Amax from the text?
When referring to "the mean area of all images as illustrated in Fig. 5b", it is not clear what exactly this refers to. The text and Fig 5b are not sufficient to precisely define Dmin.
"Dreff" (L 232) should be Deff?
What is Dv (not clear from Fig 5b)?
Dmin is likely over-estimating the thickness of Planar crystal particles, as it is difficult to get the thickness from the images. The examples Planar crystal shown in Fig 5a is likely not completely planar but features some 3D structure coming out of a dominant plane.

**Some other issues:**
=================

The units of c (L270) are not given.

Fig 8b contains the mean density from "manual measurements". This is not mentioned in the text of Sect 4.4, but likely refers to Eq. (16). Note, that this is not the same as the average density from Eq (12), see my comments regarding Eq (10)-(13).

L344 in Conclusions lists "an estimate of the speed with which the particle thickness on the plate diminishes as it melts" as one of the measured variables without saying that it is the melting speed. This text is not adequate for describing melting speed. Melting speed is h divided by melting time, not the speed at which the thickness h diminishes as that may vary with time.

Fig A1. Plotted times only, not products of times and temperatures ($\Delta T\_melt \Delta t\_melt$ and $\Delta T\_evap \Delta t\_evap$), as should be done according to Eq (A3)?

Fig B1 shows example images for aggregates. They seem to be cut (not completely imaged). Are these aggregates cut, or is this an effect of cropping when several aggregates are close to each other? It would be good to show examples with the whole hotplate indicating if several particles are observed. Are multiple particles observed falling onto the hotplate closely to each other (observed by SLR imaging) and then thermally imaged?

---

## Author Comment (AC1)

**Responses to Reviewer 1**

We thank the reviewer for the comments and review.

**Comments:** In Section 2.3 the authors present the equations that are used to estimate snowpack quantities, such as SWE and SLR. I think it would be good to explicitly point out that DEID measures falling snow properties and the observed liquid equivalent precipitation rate and bulk-snow hydrometeor density are proxies for the SWE and SLR. There are processes that take place at the surface that are neglected in the presented equations. You discuss this on page 20, but it would be good to also explicitly mention it here.

**Response:** We agree with the reviewer. We have changed the document accordingly.

**Added in manuscript [L-165] in section 2.3.** Note that the bulk density of a fresh snowpack and the height of snowpack can differ from the average density of individual snowflakes ($\overline{\rho}_{MS}$) and $H$, respectively, because snowflake settling and compaction on the ground depend on considerations such as their settling characteristics, fall angle, wind speed, the structure of snowflakes, and ambient temperature. We do not account for these processes in the calculation of the volume of freshly fallen snow layers as the impacts are largely unknown. Hence, these variables are proxies for those in the actual snowpack.

---

## Author Comment (AC2)

**Responses to Reviewer 2**

We thank the reviewer for their efforts. Clearly the reviewer spent a lot of time evaluating our paper, and all the comments helped improve the explanation and writing of the manuscript.

**Comment 1:** Eq. (1) --- Issues and questions. The term representing the heat required to increase the temperature is m*Cw*T_p,max. The specific heat of water Cw is different for ice and for liquid water. This fact seems completely neglected. No value for Cw is given.

**Response 1:** These terms were all included in the analysis, but not explicitly written in the paper. We have added all energy terms in detail and explained each term in the revision.

**Added to manuscript: [L 55-80]** The mass *m* of an individual snowflake is obtained by applying energy conservation to a control volume surrounding each hydrometeor on the hotplate. The heat gained by a snowflake from the heated plate is assumed to be equivalent to the heat required to increase the snowflake's internal energy plus the heat lost during melting and evaporation as described by the following equation

$$mC_{ice}(T_0 - T_{ice}) + mL_f + mC_{liq}(T_p - T_0) + mL_v = \kappa \int_0^{\Delta t_{evap}} (T_p - T_h(t)) A(t)dt \dots (R1)$$

Here, m is the mass of an individual frozen hydrometeor, $C_{ice}$ is the specific heat of ice, $T_{ice}$ is initial temperature of a frozen hydrometer, $T_0$ =0 °C, $L_f$ = 3.34 x $10^5$ Jkg$^{-1}$ is the latent heat of fusion of water, and $L_v$= 2.32$\pm$0.02 x $10^6$ Jkg$^{-1}$ is the latent heat of vaporization of water (see Appendix A1), $C_{liq}$ = 4.18 x $10^3$ Jkg$^{-1}$K$^{-1}$ is the specific heat of liquid water, A(t) is the area of each frozen hydrometeor and water droplet at time t, $T_p$ is the surface temperature of the hotplate during evaporation and it is constant with time, and $T_h(t)$ is the temperature of the frozen hydrometeor and/or water droplet at time t on the hotplate. $\Delta t_{evap}$ is the time required to melt and evaporate a hydrometeor, and $\kappa$ is an empirical device-specific calibration coefficient (related to the amount of heat that passes through the metal hotplate to individual hydrometeors per unit time through a unit area with a temperature gradient of one degree) determined to 7.01 $\pm$ 0.01 x $10^3$ W m$^{-2}$ K$^{-1}$ (equivalent to (k/d)_eff in Singh et al. 2021), which is independent of particle size and environmental conditions. In Eq. (1), the area of a hydrometer at any time t can be written as $A(t) = A_{ice}(t) + A_{liq}(t)$, where $A_{ice}(t)$ is the unmelted area of a hydrometeor at time t and $A_{liq}(t)$ is the melted area of a hydrometer at time t. After substituting $A(t)$ in , Eq. (1), Eq. (1) may be re-written as

$$mC_{ice}(T_0 - T_{ice}) + mL_f + mC_{liq}(T_p - T_0) + mL_v = \kappa \int_0^{\Delta t_{melt}} (T_p - T_{ice}(t)) A_{ice}(t)dt +$$
$$\kappa \int_{t_0}^{\Delta t_{evap}} (T_p - T_h(t)) A_{liq}(t)dt \dots (R2)$$

Here, t=0 corresponds to when a frozen hydrometeor hits the hotplate and t=t$_0$ is when the thermal camera sees the liquid portion for the first time. t$_0$ is the time lag between the melting and evaporation start times, respectively, which is about 0.1 s for the laboratory ice particles tested and negligible for snowflakes observed using our typical thermal-camera recording frame rates.

Note that the thermal camera can be adjusted to selectively `see' particles on the hotplate in specific temperature ranges (see section 3). To more easily evaluate Eq. (2)), the camera is set to only see hydrometeors on the hotplate after melting. Hence, $A_{ice}(t) = 0$ for $T_h(t) < 0°C$ and $\kappa \int_0^{\Delta t_{melt}} (T_p - T_{ice}(t)) A_{ice}(t)dt = 0$, which is equal to $mC_{ice}(T_0 - T_{ice}) + mL_f$. This trick allows us to remove both terms from Eq. (2), which yields

$$m = \frac{\kappa}{(C_{liq}T_p + L_v)} \int_0^{\Delta t_{evap}} (T_p - T_w(t)) \, A(t)dt \quad \ldots\ldots\ldots (R3)$$

Here, the lower integration bound is rest so that $t_0 = 0$ such that Eq (3) is evaluated for liquid phase only. $T_w(t)$ is the water droplet temperature at time t.

**Comment 2:** Further, I would expect a temperature difference in this term, the amount by which the hydrometeor's temperature is increased. Instead there is T_p,max, which is explained as "maximum surface temperature of the hotplate". It is unclear what T_p,max is and why it is used in this term.

**Response 2:** Based on the thermal camera setting, all frozen hydrometeors have initial temperature is 0 °C and it reaches maximum temperature $T_p$. Hence, temperature difference for all frozen hydrometeors during the measurement period (evaporation) is $T_p$.

**Added to Manuscript: [L -80]** Note that the initial and final temperature of all frozen hydrometeors is $T_0 = 0°C$ and $T_p$ respectively during evaporation.

**Comments 3:** Assuming that T_p,max is the temperature difference by which the hydrometeor's temperature is increased, then this implies or assumes that the whole hydrometeor (m) actually increases by this amount. Is this assumption ok, i.e. does most of the evaporation (and consequential decrease in m) happen when the melted hydrometeor reaches its highest temperature?

**Response 3:** To address this issue, a time series of the temperature of a frozen hydrometeor going through its evaporation process is shown in Fig. R1. It shows that typically, ~ 98 % of the evaporation happens at the highest temperature. And, a similar assumption was made during the calibration of k/d_eff, which reduces uncertainties.

**Added in Manuscript:[L-80]** … and more than ~98% of the mass evaporates at the highest temperature (~ $T_p$). This is illustrated in the time series of the temperature of an evaporating hydrometeor given in Fig. R1

[Figure]

Figure R1. Time series of the temperature of the melted portion of the ice particle after being placed on the hotplate. 60 μL water droplet volume was used to make ice particle for this test.

**Comment 4:** The temperature $T_w(t)$ is the "temperature of the water droplet at time t", whereas area A(t) is the "area of each snowflake at time t". Why does one refer to a droplet and the other to a snowflake?

**Response 4:** Thank you. It has been corrected.

**Comment 5:** How is "area" A defined?

**Response 5:** Area (A) was measured by counting pixels on the hotplate after **melting** and a detailed explanation is given in Singh et al. (2021). Note that the difference in area measured before and after melting of a hydrometeor is less than 3% as illustrated in Fig. 2a. We compute the area before melting by setting the temperature range of the thermal camera to $[-40,0]°C$ in post-processing. Note that when the temperature range is $[-40,0]°C$, un-melted hydrometeors have a good contrast against the hotplate (background) and allow us to easily count pixels.

**Comment 6:** The equation includes melting and evaporation. This is clear only from Leqv = Lv + Lf being the sum of the latent heats of fusion and vaporization. The integral in the equation is from time t=0 to $\Delta$ t_evap, "the time required to evaporate the water droplet". It is unclear if melting is neglected in this integral or if melting starts at t=0, i.e. if the integral includes melting and evaporation. Does evaporation also start at t=0 or does it start when melting is finished (at t = $\Delta$ t_melt)?

**Response 6:** This has been explained above in **Comment 1,** which states that we have not neglected melting. Here is one example: As you can see in Fig. 2a, the time series of ice area before and after melting start at t ~ 0. In Fig 2a, observation of evaporation starts at t = 0.1s (based on the thermal camera recording speed 10 fps). At t =0, the area of ice is ~ 2 cm$^2$, and area of melted ice is ~ 0 cm$^2$. At t= ~ 4 s, area of ice is ~ 0 cm$^2$, and area of melted ice is ~ 2 cm$^2$. Evaporation started at ~ 0 s. At t = $\Delta t_{melt}$ (~4 s), melted ice reached at ~ $T_p$ (Fig. 2b).

**Added in Manuscript:[caption figure 2]:** The lag between melting and evaporation start time is 0.1 s, which is equal to the thermal camera sampling period.

**Comments 7:** $\kappa$ is determined by evaporating droplets of known mass (in Singh et al. 2021, water droplets of 20μL). The "empirical calibration coefficient" $\kappa$ seems to represent the specific heat capacity of the contact area between hotplate and melted hydrometeor. $\kappa$ is (k/d)_eff in Singh et al. 2021, who call this the calibrated value of the term k_w/d_w in their equation for R2 (in text; for R2 see their Fig 2 and Eq 15) without mentioning anywhere what k_w and d_w are.

**Response 7:** $\kappa$ is the amount of heat that passes through a metal plate to water droplets per unit of time through a unit area with a temperature gradient of one degree. Singh et al. 2021, in Eq. (15), R1 and R2 are the thermal resistance across the aluminum plate and water droplet, respectively, where $k_{AL}$ and $d_{AL}$ are thermal conductivity and thickness of the aluminum plate. Similarly, $k_w$ and $d_w$ are parameters for water droplets on the aluminum plate but values of $k_w$ and $d_w$ are not known. Hence, we calibrated k/d_eff values.

**Added in Manuscript: [L 65].** The amount of heat that passes through a metal plate to water droplets per unit of time through a unit area with a temperature gradient of one degree

**Comment 8:** $\kappa$ is determined only for liquid droplets and only for one size. Is $\kappa$ a constant that can be applied to frozen and liquid hydrometeors as done in Eq. (1)?
Melting neglected in Eq. (1)??
Initially, at deposition at t=0, the contact is at a few points only. Most likely it takes a very short time for a liquid layer to form at the hotplate, which provides the contact area needed to apply Eq. (1).

**Response 8:** $\kappa$ is a constant independent of particle size and environmental conditions added in manuscript line 65. and $\kappa$ is an empirical device-specific calibration coefficient (related to the amount of heat that passes through the metal hotplate to individual hydrometeors per unit time through a unit area with a temperature gradient of one degree). As discussed in Eq. (3) above, Eq. (3) only works for frozen and liquid hydrometeors after melting. Therefore, $\kappa$ value works for both frozen and liquid

hydrometeors. We agree that the melting process is very short and area of ice between deposition at t =0 and t= $\Delta t_{evap}$ are provided in Fig. 2a and only after the melting area is required in Eq. (2).

**Comment 9:** Heat lost from the hydrometeor or melted hydrometeor is neglected. This is not mentioned, motivated or discussed.

**Response 9:** Now this is discussed from line 60 to 75 in the manuscript.

**Eq. (2) --- Issues and questions**

**Comment 10:** The time varying temperature difference from Eq. (1) is replaced with the constant Δ T_evap referring to Rees et al. 2021, who should be demonstrating that this can be done. I could not see that Rees et al. 2021 motivate this in any way, they simply do the same substitution from one equation to the next. As with Eq. (1), it is unclear if the integral starting at t=0 includes melting or only shows evaporation. In the former case, Fig2b shows that the temperature difference cannot be approximated as constant. In the latter case, the integral is over the time range in white in Fig2b and the temperature difference Δ T_evap may be approximated as constant.

**Response 10:** The mass of hydrometeors is calculated using time varying temperature difference Eq. (3)

$$m = \frac{\kappa}{\left(C_{liq}T_p + L_v\right)} \int_0^{\Delta t_{evap}} \left(T_p - T_w(t)\right) A(t)dt \quad \dots\dots\dots\dots (R4)$$

and mean temperature difference

$$m = \frac{\kappa}{\left(C_{liq}T_p + L_v\right)} \Delta T_{evap} \int_0^{\Delta t_{evap}} A(t)dt \quad \dots\dots\dots\dots (R5)$$

The mass from both equations matches well, within 2 % error, which is within uncertainty measurement in mass (Note that for calibration of $\kappa$ in Eq. (R5)) $\Delta T_{evap}$ was used and is a consistent method to reduce uncertainty in the measurements). Where, $(\Delta T_{evap} = \overline{\left(T_p - T_{liq}(t)\right)})$. As we discussed earlier, Eq. (R3) only required evaporation parameters and measured mass using time varying temperature difference and mean temperature difference for the Fig 2 dataset and the results show 1.6 % error. Regardless, we are not using above approximation for calculation of mass and density. We have removed the Eq. (2). Note that Fig 2 b shows minimum temperature of ice particle during melting.

**Comment 11:** The "particle's effective thickness in the direction normal to the hotplate h" seems to be defined as the height of the quasi hemispherical "droplet" on the hotplate (Fig1). Why is this called "effective" thickness? The melting speed relates $h$ and the time required to "melt an individual snowflake". How is h defined for a "frozen hydrometeor"?

**Response 11:** Sorry for this confusion. In Fig1a, $h$ was the maximum thickness of the hemispherical ice particle, which has been replaced with radius (R) in Fig 1a. The effective thickness of hemispherical ice particles and snowflakes is defined as Volume = Area x effective thickness ($A_p$ $h$). In the case of a hemisphere, V ($\frac{2\pi}{3}R^3$)= A($\pi R^2$) x (2R/3), where $h = 2R/3$ is effective thickness of hemisphere. We have modified Fig.1a. The effective thickness of frozen hydrometeors is illustrated in Fig1a,b. The method is also explained in the text on lines 120-135.

[Figure]

(a)

$$\kappa \int_{o}^{\Delta t_{\text{melt}}} A(t)(T_p(t) - T_s(t))dt = mL_{ff}$$

$$A(t = 0) = A_p$$
$$h = {}^{2R}/_3$$

$$v_{\text{melt}} = \frac{h}{\Delta t_{\text{melt}}} = c_{\text{melt}} \Delta T_{\text{melt}} \quad \text{at all environmental conditions}$$

(b)

[Figure]

$$V_s = \sum_{ij} V_{ij} = \sum_{ij} A_{ij} h_{ij} \quad --- (1)$$

$$V_s = c_{melt} \sum_{ij} A_{ij} \Delta T_{ij} \Delta t_{ij} \quad --- (2)$$

$$V_s \approx c_{melt} A_p \Delta T_{\text{melt}} \Delta t_{\text{melt}} \quad --- (3)$$

Figure R2 (a) Schematic of heat transfer from an aluminum hotplate to a solid hydrometeor during melting. R is the radius, and $A_p$ is the base area of hydrometeor on the hotplate. h is the effective thickness of the hemisphere that is 2R/3, For the methodology presented here, a control volume is defined to wrap around the hydrometeor. (b) Schematic of calculation of volume, area, and height of frozen hydrometeors on the hotplate.

**Comment 12:** Eq. (3) implies that the "volume of a snowflake" is Ap * h.
When P_0 is later determined (Sects 3.1/3.2 and Sect 4.1), frozen quasi-hemispherical solid ice pellets are used. As an example, one size is generated from water droplets of 60μL volume and stated to have a measured height h of 1.95mm and area A (why is this not called not Ap?) of 0.502 cm2 (L278-279). The volume of these ice pellets is thus (according to Ap * h) 98μL. This cannot be explained by the lower ice density andd shows that Ap * h is not the volume of these ice particles. However, this volume is what Eq. (3) is based on. This then affects the following equations too, in particular Eqs (7) and (8), which are used to determine the density of a frozen hydrometeor.

**Response 12:** Here, $A$ is replaced by $A_p$. The volume of the hemisphere $= \frac{2\pi}{3} R^3 = A \frac{2R}{3} = 0.502$ (2*1.95/3) = 65 μL. Also, these values were for ice particles after melting, which is corrected on Lines 190-195, and Table 1 has been added to section 3.1 where the area and height of each sample of ice particles are included.

**Comment 13: Eq (4) --- Issues and questions**

 L_ff should be L_f?

The equation refers only to melting. It seems to implicitly approximate the temperature difference between hotplate and frozen part of hydrometeor as constant Δ T_melt. Is this justified? I am not sure when looking at Fig2b (the grey part).

What happens to the integral? Why have an integral over A over a constant 1 and not write out what that is? How you get to that integral is also unclear. There seem to be implicit assumption that are not mentioned.

**Response 13:** $L_f$ is the latent heat of fusion of ice, while $L_{ff}$ is the total energy required to melt ice particles, $L_{ff} = mC_{ice}(T_0 - T_{ice}) + mL_f$; this has been added to the manuscript on Line 105. We agree that the temperature difference between the hotplate and the frozen part of the hydrometeor is not constant and varies with time. We address this in the response to **Comment 10**. Results show within 2% uncertainty between the time series of temperature difference and mean value of time series of temperature difference $(\Delta T_{melt} = \overline{\left(T_p - T_{ice}(t)\right)})$ .

**Added in Manuscript**: We define $\Delta T_{melt} = \overline{\left(T_p - T_{ice}(t)\right)}$ Where t varies from 0 to $\Delta t_{melt}$. We have re-defined melting velocity using temperature differences across hotplate and frozen hydrometeor in lines 105-115. Additional information for calculation of volume of frozen hydrometeor is provided in Line 125-130 using $\Delta T_{melt}$.

**Comment 14:** Eq (5) --- Issues and questions

The concept of "the 'effective mechanical work rate' or mechanical power ... required to melt a snowflake" is not adequately introduced, in particular the "constant 'melting pressure'" P_0. Apparently P_0 was found to be constant empirically, and these equations (Eq.5) represent the attempt to derive that with some integrals that seem mathematically dubious. It is unclear what integrals over V_s or A*h between t=0 and Δ t_melt as it is mathematically wrong. As with Eq. (4), there seem to be implicit assumptions not mentioned, and the unresolved integral over A appears.

**Response 14:** This work interpretation has been removed in favor of a straightforward empirical description, and the manuscript has been updated. In addition, we carefully looked through the integrals to make sure they have been corrected. Thank you for the keen eye!

**Added to manuscript:** [L 105-135]. We propose a method to measure $v_{melt}$ as a function of the temperature difference across a hydrometeor $(\Delta T_{melt})$ -- and hence the heat-transfer rate -- as illustrated in Fig. 1a. During the time it takes to melt a snowflake, $\Delta t_{melt}$, a hydrometeor receives a quantity of energy equal to $mL_{ff} = mC_{ice}(T_0 - T_{ice}) + mL_f$ (that is, the sum of the internal energy per unit mass of a frozen hydrometeor and its latent heat of fusion) from the hotplate, which is independent of density of frozen hydrometeors. $v_{melt}$ is associated with conductive heat flux rate ($\kappa$ $\Delta T_{melt}$) from the hotplate to the frozen hydrometeors during melting. We hypothesize that $v_{melt}$ as a function of the temperature difference across a hydrometeor $(\Delta T_{melt})$, and we may write the $v_{melt}$ as

$$v_{melt} = c_{melt}\Delta T_{melt} \quad (R6)$$

where $\Delta T_{melt} = \overline{T_p - T_s(t)}$ during the melting process and $T_s(t)$ is the surface temperature of the frozen portion of the particle during melting and $c_{melt}$ is a constant determined experimentally (see section 4.1) Now, if $v_{melt}$, from Eq. (4) is substituted into Eq. (3) the MS density equation can now be written as

$$\rho_{MS} = \frac{m}{c_{melt}A_p\Delta T_{melt}\Delta t_{melt}} \quad (R7)$$

The melting parameter $\Delta t_{melt}$ is quite short for low-density snowflakes, and hence requires high-frequency recording of thermal images resulting in the generation of a tremendous amount of data, which is not convenient for field experiments.

In field experiments, the DEID measures $\Delta t_{evap}$, which is much longer than $\Delta t_{melt}$. Fortunately, a relation between $\Delta t_{melt}$ and $\Delta t_{evap}$ can be easily derived using experimental data (see Appendix A). By estimating the average conductive heat-transfer rate during the melting and evaporation processes, we may substitute $\Delta T_{melt}\,\Delta t_{melt} \approx (L_{ff}/L_{vv})\,\Delta T_{evap}\Delta t_{evap}$ into Eq. (5), which yields

$$\rho_{MS} = c\,\frac{m}{A_p\Delta T_{evap}\Delta t_{evap}}\,(R8)$$

Here, the constant c is given by

$$c = \frac{L_{ff}}{L_{vv}c_{melt}}\,(Ksm^{-1})$$

which is derived from a combination of thermodynamic and laboratory calibration constants that must be determined experimentally (see section 4.1).

In practice, Eq. (6) is evaluated following the methodology shown in Fig. 1b. Snowflakes of complex shapes do not have simple height relationships like h = 2R/3 as shown in Fig. 1a. Hence, a method is required to determine the height and volume of every pixel within a snowflake. If $A_{ij}$ is taken as the area of $ij^{th}$ pixel, $h_{ij}$ the height of a frozen hydrometeor normal to the hotplate that is associated with the $ij^{th}$ pixel, and $V_{ij}$ is the volume of $ij^{th}$ pixel, then we may write

$$V_s = \sum_{ij} V_{ij} = \sum_{ij} A_{ij}h_{ij} = c_{melt}\sum_{ij} A_{ij}\Delta T_{ij}\Delta t_{ij}$$

which can then be estimated using the MS method on a pixel by pixel basis. Here, $\Delta T_{ij}$ is the temporal mean temperature difference across $ij^{th}$ pixel, $\Delta t_{ij}$ is the melting time of the $ij^{th}$ pixel and $c_{melt}$ is calibration constant of melting velocity. This study used Eq. (9) to calibrate laboratory ice particles and compare snowflake habits. For field observations, we make the following simplification to determine the volume

$$V_s = c_{melt}A_p\frac{1}{n}\sum_{ij}\Delta T_{ij}\Delta t_{ij} \approx c_{melt}A_p\Delta t_{melt}\frac{1}{n}\sum_{ij}\Delta T_{ij} = c_{melt}A_p\Delta t_{melt}\Delta T_{melt}$$
$$= \frac{1}{c}A_p\Delta t_{evap}\Delta T_{evap}$$

Here, $A_p = nA_{ij}$ (assuming all pixels have the same area), n is the total number of a pixels associated with a frozen hydrometer, $\Delta T_{melt} = \frac{1}{n}\sum_{ij}\Delta T_{ij}$ is the spatial and temporal mean temperature difference across a frozen hydrometer during melting, $\Delta t_{melt} \approx \Delta t_{ij}$ is the melting time of a frozen hydrometer and $c = \frac{L_{ff}}{L_{vv}c_{melt}}$.

**Comment 15:** Eq (6)-(9) --- Issues and questions

These equations follow from the above equations, i.e. they are based on the assumptions and approximations of those so that it is difficult to see under which conditions they apply and/or how the density rho_MS is implicitly defined. See also the issues regarding Eq (3) about interpretation of volume and definition of h.

One assumption used to derive Eq. (8) from Eq (7) is the substation shown in L 108 to replace $\Delta$ T_melt and $\Delta$ t_melt with $\Delta$ T_evap and $\Delta$ t_evap.
This is based on Eq (4)/Eq (A1) and Eq (A2) with all their assumptions and issues.

**Response 15:** Please see the response to **Comment 13** about Eq. (6-9). We have plotted associated with T_melt x $\Delta$ t_melt vs $\Delta$ T_evap x $\Delta$ t_evap for ice particles and snowflakes in Appendix A2 (Fig. R3 ), which shows an approximately similar relation between melting and evaporation parameters.

[Figure]

Figure R3. Experimentally measured $\Delta T_{melt}\,\Delta t_{melt}\;$ and $\;\Delta T_{evap}\,\Delta t_{evap}\;$ for laboratory ice particles. The slope of the line is quite close to $L_{vv}/L_{ff}$. An error bar is associated with ten samples in each ice particle size.

**Comment 16  Eq (10)-(13) --- Issues and questions:** The accumulated snow water equivalent SWE is simply the snow water equivalent rate of precipitation (from Eq. 10) times $\Delta$ t_res. Why use an integral (in Eq 11) if all that is involved is a single value of $\Delta$ m and one sampling interval $\Delta$ t_res?

**Response 16:** We agree, this has been modified as $SWE = \ \dot{SWE}$ x $\Delta t_{res}$

**Comments 17:** Eq. (12) defines an average hydrometeor density of an ensemble of hydrometeors as the sum of their masses divided by the sum of their estimated volumes. I do not agree with calling this "average density of a freshly fallen snowpack layer". If it was the average snowpack density, then Eq. (13) describes the snow accumulation rate, as claimed. But given the definition of this average density, H from Eq. (13) is the height that one would get if all hydrometeors pack exactly (sum of volumes of hydrometeors divided by the surface over which they were collected), i.e. without effectively neither leaving any space in between them nor overlapping.
In other words, I don't agree that one can generally determine the height of the snowpack and the

accumulation rate from the volumes of the individual hydrometeors without any knowledge of how efficiently they pack.

**Response 17:** We agree with the reviewer and it has been mentioned in Lines 325-330, "The bulk density of a fresh snowpack can differ from the average density of individual snowflakes and the 1-min average because snowflake settling and compaction on the ground depends on such considerations as their settling characteristics, fall angle, wind speed, the structure of snowflakes, and the ambient temperature. We do not account for these processes in the calculation of the volume of freshly fallen snow layers as the impacts are largely unknown." We also added Lines 155: by assuming neither leaving any space between snowflakes nor overlapping.

**Comment 18 - DEID laboratory set-up:** The DEID is described briefly here, the previous publication about the DEID and its method (Singh et al. 2021) should contain the details.
The set-up, however, is flexible and in this Manuscript is described as a hotplate of 9 cm x 6 cm that is thermally imaged using 531 pixels x  362 pixels of the thermal camera yielding a "spatial resolution" of 0.2 mm/pixel. It is, without mentioning this, assumed that the thermal imagery is looking vertically down at the hotplate. What is not mentioned here, neither for the set-up in Singh et al. 2021, is the actual inclination of around 30 degrees (see Fig 3c). Does this, and if so to what extent, affect the determination of Ap and A?

**Response 18:**  Due to the inclination of the thermal camera, the maximum error in area measurement is 1.6%, and it is corrected using the dewarping function.

**Added to manuscript:** [L180-185]. The thermal camera was looking vertically downward at the hotplate at an angle. As a result, the maximum error in area measurement is 1.6%, and the error in area was corrected using the dewarping function.

**Comments 19**: Fig 3b shows a "Side-view of a surface temperature contour plot of an ice particle obtained using the thermal camera." Has this been obtained with the configuration shown in Fig 3c?

**Response 19:** Yes.

**Added in Manuscript: (figure caption)** … using temperature range in thermal camera [-40,0]°C

**Comment 20 - Sample preparation:** L 164-166 describe the sample preparation to manufacture ice particles for laboratory validation. In a freezer, one of eight different known volumes is applied, using a micropipette, to "a flat silicone mold". Is this a flat silicone sheet rather than a mould suggesting a specific shape? The formed ice particles are referred to as "hemispherical". I guess they will be close to, but not exactly hemispherical, as can be seen, for example, from the value of Ap of 0.502 cm2 (L278- 279) for ice particles produced from 60μL of liquid water (a hemisphere of liquid water would have 0.29 cm2). The shape of these sample ice particles should be better described. See also comments on volume and h regarding Eq. (3).

**Response 20:** The shape, volume, area, and height of the sample of laboratory-made ice particles are added in Table 1. The only cross-sectional area of each ice particle sample was maintained using a base silicon mold while making ice particles. The value of $A_p$ and maximum thickness $h$ (L 278- 279) for ice particles produced from 60μL of liquid water is corrected, which was listed for liquid water after melting.

**Table 1.** The shape of sample ice particles made in a laboratory: $V_{\text{pipette}}$ is the volume of water applied by the pipette, $A_p$ is the cross-sectional area of ice particles measured by the thermal camera on the hotplate, and $h$ is the maximum thickness of ice particles determined from the laser pointer system.

| $V_{\text{pipette}}$ ($\mu$L) | $A_p$ (mm$^2$) | $h$ ($mm$) |
|---|---|---|
| 5 | $5.65 \pm 0.09$ | $1.21 \pm 0.07$ |
| 10 | $9.10 \pm 0.09$ | $1.64 \pm 0.03$ |
| 20 | $13.90 \pm 0.11$ | $2.23 \pm 0.08$ |
| 30 | $17.60 \pm 0.09$ | $2.54 \pm 0.03$ |
| 40 | $22.61 \pm 0.11$ | $2.28 \pm 0.07$ |
| 50 | $26.81 \pm 0.17$ | $2.85 \pm 0.05$ |
| 60 | $31.13 \pm 0.16$ | $3.25 \pm 0.09$ |
| 70 | $34.56 \pm 0.14$ | $3.34 \pm 0.08$ |

**Comment 21:** Sect 3.2/Sect 4.1: what is the ambient temperature in the lab when testing environmental variability? In field evaluations (Sect 4.3) the ambient temperature varies. Do results depend on ambient temperature, which for calibrations was plus 18 degree C?

**Response 21:** The calibration constant $\kappa$, mass m, and Area $A_p$ are independent of environmental variability (Singh et al. 2021), while $\Delta t_{evap}, \Delta t_{melt}, v_{melt}, \Delta T_{melt}, \Delta T_{evap}$ parameters depend on environmental variability, and all the listed terms are measured directly during experiments. The laboratory calibration of $v_{melt}$ was also performed at the different environmental conditions and $v_{melt}$ was only the function of $\Delta T_{melt}$, which we measured directly.

**Comment 22:** L 274: "Here, Vpipette is volume of the ice particle created by pipetting a known volume of water." Should Vpipette not be the volume of liquid water dispensed from the pipette to generate an ice particle?

**Response 22:** We agree that Vpipette is the water droplet volume, not ice particle volume; This has been corrected.

**Corrected in manuscript [L 305]** … here, $V_{\text{pipette}}$ is the volume of a water droplet, not the ice particle volume.

**Comment 23:** Sect. 4.1 last sentence (L 283-285): The "80 ice particles with different shapes and sizes" were presumably all of the same size and same shape. I am assuming this as it is in the same paragraph as the description of test particles as produced from 60μL of liquid water. Else, what does "different shapes and sizes" refer to?

Tab 3: What is "Percentage of salt"?

**Response 23:** A different unknown shape was made by pipetting water droplets onto random surfaces with varying contact angles.

**Added in manuscript [L 315] …** and was made by pipetting water droplets on arbitrary surfaces with different contact angles. The percentage of salt and water added is in Table 4.

**Table 4.** The density of frozen salt-water particles measured using the MS method and determined theoretically based on a salt-water ratio by weight (percentage of salt and water).

| Percentage of salt (%) | Percentage of water(%) | $\rho_{\mathrm{MS}}$ (kg m$^{-3}$) | $\rho_{\mathrm{theoretical}}$ (kg m$^{-3}$) |
|---|---|---|---|
| 1.47 | 98.52 | $969 \pm 37$ | 1008 |
| 2.75 | 97.24 | $988 \pm 26$ | 1015 |
| 5.07 | 94.92 | $1002 \pm 38$ | 1028 |
| 7.16 | 92.83 | $1018 \pm 24$ | 1040 |
| 8.86 | 91.13 | $1028 \pm 31$ | 1050 |

**Comment 24: Uncertainties** - There is no discussion of how measurement uncertainties affect the measurements of mass and density.

Errors in $A$ as determined by thermal imagery?

Micropipette accuracy? L 152: "1.00/1.20 %/µL", what does that mean? 1% or 1% for each 1.2 µL; 1% or 1.2 µL (when 1% and when 1.2 µL)?

Accuracy of measurements of $h$? Accuracy of temperature measurements?

Maybe less important: accuracy of time measurements? With 15 fps I guess error in time is on the order of 1/15 s.

**Response 24:** Systematic and Random error analysis added in Appendix A: Direct measurements made by the DEID include snowflake area, temperature, and evaporation time, for which the respective uncertainties are 1.4%, 0.3%, and 1.6%, respectively. Due to the inclination of the thermal camera, maximum error in area measurement is 1.6%, and it is corrected using Matlab's dewarping function. The uncertainties in derived quantities (using a standard propagation of uncertainty analysis) such as mass (m), height (h) and density ($\rho_{MS}$), are 4.3%, 2.9% and 8.6% respectively.

Micropipette accuracy updated in **L 180**: accuracy of 1\% and maximum accuracy 1.2 $\mu L$ at highest volume.

**Comment 25 - Sect 3.5 --- geometrical volume estimates:** The explanation of how volume is determined is not sufficient. The quantities Dmin, Dmax, A, mean(A), and Dv are not defined adequately. Planar crystals are approximated as a disc. Is this disc a cylinder (as suggested by Fig 5b) or a column with base Amax (as suggested by the definition of Amax from the text?

When referring to "the mean area of all images as illustrated in Fig. 5b", it is not clear what exactly this refers to. The text and Fig 5b are not sufficient to precisely define Dmin. "Dreff" (L 232) should be Deff?
What is Dv (not clear from Fig 5b)?

Dmin is likely over-estimating the thickness of Planar crystal particles, as it is difficult to get the thickness from the images. The examples Planar crystal shown in Fig 5a is likely not completely planar but features some 3D structure coming out of a dominant plane.

**Response 25:** Here, planar crystals are approximated as a disc, and the volume of the disc is $\pi R^2 h$, where $h << R$, which is distinct from a cylinder or column.

**Added in Manuscript [L 260-265]:** $D_{min}$ is the minimum dimension (representing the thickness of the disc) in the 2-D plane as indicated in Fig. 5a column I row V. The volume of graupel was estimated as a sphere $V = \frac{\pi}{6} D_{Reff}^3$, where $D_{Reff}$ is the effective circular diameter estimated using the SLR camera imaging the snowflake falling in the air and estimated as $D_{Reff} = \sqrt{\frac{4}{\pi} \overline{A}}$. $\overline{A}$ is the two-dimensional area of all images as illustrated in Fig. 5a, column II.

**Comment 26:** The units of c (L270) are not given.

**Response26:** The units of $c$ have been added on L 115.

**Comment 27:** Fig 8b contains the mean density from "manual measurements". This is not mentioned in the text of Sect 4.4, but likely refers to Eq. (16). Note, that this is not the same as the average density from Eq (12), see my comments regarding Eq (10)-(13).

**Response 27:** Added in the Fig 8 caption. Note that the MS method and manual measurement are compared without knowing the information about snowpack, as we mentioned earlier.

**Comments 28:** L344 in Conclusions lists "an estimate of the speed with which the particle thickness on the plate diminishes as it melts" as one of the measured variables without saying that it is the melting speed. This text is not adequate for describing melting speed. Melting speed is h divided by melting time, not the speed at which the thickness h diminishes as that may vary with time.

**Response 28:** We agree with reviewer and have corrected this in the conclusion that melting speed is $h$ divided by the melting time.

**Comment 29:** Fig A1. Plotted times only, not products of times and temperatures ($\Delta$ T_melt $\Delta$ t_melt and $\Delta$ T_evap $\Delta$ t_evap), as should be done according to Eq (A3)?

**Response 29:** We have corrected this and plotted Fig. A1 (Fig. R3), the product of $\Delta$ T_melt $\Delta$ t_melt and $\Delta$ T_evap $\Delta$ t_evap.

**Comments 30:** Fig B1 shows example images for aggregates. They seem to be cut (not completely imaged). Are these aggregates cut, or is this an effect of cropping when several aggregates are close to each other? It would be good to show examples with the whole hotplate indicating if several particles are observed. Are multiple particles observed falling onto the hotplate closely to each other (observed by SLR imaging) and then thermally imaged?

**Response 30:** The image shown in Fig B1 is a cropped aggregate snowflake on the hotplate just after it has melted. We have also added an entire hotplate image with snowflakes.

[Figure]

Figure R4. (a) Black and white binary thermal images of aggregate type of snowflakes in different stages of melting and evaporation on the DEID heated plate observed at Alta. (b) Cropped aggregate snowflake images on the hotplate just after melting.

The overlap is minimized by the laser-sheet thickness (~ 5 cm) and the SLR camera setting (160 $\mu m$ pixel$^{-1}$). The probability of coincidence on the hotplate depends on the precipitation rate, and negligible overlap was observed for a precipitation rate of ~ 1 mm/hr, and a maximum of 4.9% coincidence probability was observed during the highest SWE rate 15.6 mm/hr, illustrated in Fig R5.

[Figure]

Figure R5. Probability of coincidence as a function of SWE rate.

---

## Referee Report (RR1)

Below are comments based on my second review of manuscript amt-2023-148, i.e. the review of the revised version amt-2023-148-manuscript-version2.
I am refering to Line, Sect., Eq., Fig., etc. numbering of amt-2023-148-ATC1 and amt-2023-148-manuscript-version2.

Some changes and improvements have been done. This has clarified several issues that I had while reading the manuscript in the first review. However, some issues such as wrong, sloppy, or confusing formulations still remain or have been introduced in new explanations. Also a discussion of the seemingly good results is missing.

**Seemingly good results:**
In my previous review I have noted that despite many assumptions, the results are seemingly good. This is both surprising and interesting and needs better and further discussion. In particular it is interesting how DEID estimated well snowpack densities agree with manually measured density despite the fact that average hydrometeor density is used as proxy of the snow pack density.

Validation measurements and calibrations (vmelt, kappa, c, cmelt) are using a limited variety of very simply-shaped hydrometeors. Is kappa the same for ice and liquid (melting and evaporation)? How/why can these then be used for real snow particles, why are the results so good? This is not obvious and should be briefly discussed.

**Area not properly defined:**
Area A(t) (and Aice and Aliq) is still not prpoperly defined. You simply state that it is the area of the ice particle, melted hydrometeor …
Formulations like "A(t) is the area of each frozen hydrometeor and water droplet at time t," are ambiguous.
This is ambiguous and may for example refer to the total surface area.
Regarding the heat transfer, it should be a contact area between hydrometeor and hot plate (or some sort of effective contact area).
Regarding your measurement method, it should be cross-sectional area as seen on your 2D images (Ap you call projected area).

**Errors in Eq. (1)-(3):**
In Eq. (1), it looks like Th(t) is the temperature of the whole hydrometeor (having a cross-sectional area A(t)).
This is a simplification hiding details that you likely consider in your algorithms. In addition to the integral over time there should be an integral over the area and Th(t), as well as Tice(t) and Tliq(t), is evaluated at each location. Presumably in your algorithm you do this double integral as two sums, one over all hydrometeor pixels of the image and one over all time steps (images) during melting and evaporation.
Without a double integral, I don't see how you can go from Eq. (1) to Eq. (2) only using A(t) = Aice(t)+Aliq(t). (In your algorithm, a pixel is either ice or liquid, thus you can split the area integral in the two parts you indicate in Eq (2). However, just as Eq. (1), Eq. (2) is wrong (over-simplified).

To say that the camera doesn't see ice, and therefore Aice=0 is wrong. If starting with ice, then of course neither is Aice zero, nor the terms m*Cice*(T0 -Tice)+m*Lf.

You seem to assume that heat transfer through Aice goes exclusively into temperature increas of ice and melting of ice (and heat transfer through Aliq does not go into these ice terms).

So I would clearly state this assumption (important since during part of the time integral in Eq. (3) ice is still present) and directly introduce Eq. (3), i.e. skipping Equations (1) and (2), making sure the integral is properly formulated. It is worth already here mentioning that when the camera is set to only see hydrometeors after melting, this is done in post-processing only.

Be consistent, e.g, if using Aliq, then use it also in Eq. (3).

Reformulte "Note that the initial and final temperature of all frozen hydrometeors is T0 = 0degC andTp, respectively, during evaporation", which is confusing as you refer to a "frozen hydrometeor" (during part of the evaporation part of the hydrometeor is frozen and below T0).

In Eq. (3) it is wrong to use Cliq*Tp. The specific heat Cliq needs a temperature difference (i.e. Tp-T0). The same applies to Equations (B1) and (B2).

**Sect 2.2 Particle density**
**Unclarities remain about h and volume estimate:**

Fig. 1 is not "illustrating" well a heat transfer rate and control volume. They need to be better explained in the text and equations.

That m*Lff is "the sum of the internal energy per unit mass of a frozen hydrometeor and its latent heat of fusion" is another example of a wrong/sloppy formulation.

What is the reason or motivation behind hypothesis Eq. (5)?

Explaining DeltaTmelt below Eq. (5), you have Tp(t). But earlier you stated that Tp was constant.

You refer to the wrong Appendix (should be B not A).

DeltaTevap is not defined.

Equation (8) is wrong.

Without your explanation in the response, one needs some guessing to understand the reasoning behind the "simple height relationships like h = 2R/3". R is (only in caption of Fig 1) introduced as "radius of the hydrometeor", h as the "effective thickness of the hemisphere". You should (more clearly) refer to the special case of a hemispherical ice particle here. You still need a general definition of h in the text. This is important for clarity of course, but also as the definition of vmelt seems to be based on h.

It is unclear how, in the laboratory, you can use Eq. (9) with its term hij. How can you determine all hij?

"...used Eq. (9) to calibrate laboratory ice particles and compare snowflake habits." What do these refer to?

Line 137: "Note that are impacted by variability..." Something is missing.

In caption of Fig. 2 "(a) Time series of the area of the ice particle (dashed black line) and the melted portion of the ice particle (solid black line)" is another example of a wrong/sloppy formulation.

**Sect 2.3 bulk snowpack-derived quantities**
The assumption "... by assuming neither leaving any space between snowflakes nor overlapping" is wrongly placed, it is not needed for the avaerage density defined by Eq. (12).
The snow accummulation rate in Eq. (13) needs the above assumption. The assumption, however, seems unmotivated and Eq. (13). Link your text "Note that the bulk..." to that assumption.

**Sect 3.1: ice partile height vs effective height**
You don't seem to refer to Fig. 3 (only to Fig. 3d).
Fig. 3b, Fig. 3d, and in text related to Fig 3d, the height h seems to be a height rather than the effective height h. Also, in Tab. 1, h seems to be close to R derived from Ap (not to 2/3*R as I would expect) if the ice particle were a hemisphere. In Fig. 3b, the ice particle doesn't look hemispherical (max height is less than R). Deff in that figure is not explained.
L. 376-377 also talk about height rather than effective height. Check for consistency.
If the side view in Fig. 3b is taken with the thermal camera shown in Fig. 3b, i.e. from above (Response 19), then I am not sure how it can be a side view.
You should be clearer and more detailed in your description of preparing ice particles in Sect.s 3 and 4 and resulting shapes and contact angles. I am wondering about the role of the silicone mold and didn't understand earlier that it has a certain shape (circular deepening or flat or other shapes deepenings as suggested by Line 315?).

**Fig. 5**
In the text related to Fig.5 explaining the geometrical volume estimates, it is unclear what Dv is (column III aggregates).

**Approximations in Appendix B**
As you equate (B1)=(B2) you need to relate Equations (B1) and (B2) to the whole mass (that is melted completely in B1 and then evaporated completely in B2). You need to specify over which part of the area A (or pixels) you effectively integrate in each of these equations (see comments on integrals in Equations (1) and (2) above).
Explain what the "averaging approximation" means. Note that you already use an approximation when using an average DeltaTmelt and DeltaTevap that should be explained (or refer to somewhere in the paper). (See your Response 10, for example)

**Appendix D errors:**
An error of only 1.4% for the area measurement seems very good. Can you describe briefly how this was determined or estimated?

Uncertainties in h: are these based on the laser measurements or the indirect method involving assumptions and approximations around DeltaTevap*Delta_tevap?

---

## Author Response (AR2)

**Responses to Reviewer 2**

We thank the reviewer for their efforts and all the comments that helped improve the paper's clarity and manuscript writing.

**Comment 1: Seemingly good results:**

**Comment 1:** In my previous review I have noted that despite many assumptions, the results are seemingly good. This is both surprising and interesting and needs better and further discussion. In particular, it is interesting how DEID estimated well snowpack densities agree with manually measured density despite the fact that average hydrometeor density is used as proxy of the snowpack density.

**Response 1:** We agree that snowpack density estimated by DEID using the average density of snowflakes is surprising. The following statements may help explain these results:

(1) The fresh snowpack minimally settles during the short interval of 12 hours between manual measurements. That is, comparisons are made with 12-hour storm boards.
(2) There may be minimal overlap of snowflakes in snowpack. That is, the sum of the heights of the snowflakes is equal to the total depth.
(3) In reality, there is overlap, but on "average," overlap is minimal.
(4) Our results imply that the density of the snowpack is equal to the average density of individual snowflakes, which means that the snowflakes "pack" with a density similar to the density of the snowflake. We note that sometimes this is not true, for example, with errors of 15%.

This discussion has been added to the paper in the Conclusions.

**Comment 2:** Validation measurements and calibrations (vmelt, kappa, c, cmelt) are using a limited variety of very simply-shaped hydrometeors. Is kappa the same for ice and liquid (melting and evaporation)? How/why can these then be used for real snow particles, why are the results so good? This is not obvious and should be briefly discussed.

**Response 2:** Indeed, kappa was calibrated in the lab with water droplets and idealized ice particles (Singh et al. 2021), and the kappa value that we found is valid for both water droplets and ice particles. To deal with complex snowflake shapes, an alternative validation method was needed. Hence, as presented in section 4.3, individual snowflake densities were measured using the SLR camera-laser system (for volume) and the DEID (to obtain mass). This method agrees well with the $v_{melt}$ densities, which indicates kappa is suitably applied to all snowflake types.

We believe that the results the high level of agreement is because all reported measurements are based on the evaporation method (liquid phase), i.e. it is based upon Eq. (8) (previously Eq. (3)). SWE measured for seventeen storms shows excellent agreement with manual measurements (see Figure 9a), with a coefficient of determination of 0.99. The implication is that that kappa is well

suited for all types of snow after melting. The parameters $v_{melt}$, c, and $c_{melt}$ all depend on kappa and $\Delta T_{melt}$ . Hence, we can use $v_{melt}$, kappa, c, and $c_{melt}$ for all types of snowflakes. The results are promising because a similar assumption/uncertainty (e.g. error in temperature, area, time, temporal and spatial averaging) was made during kappa calibration.

**Area not properly defined:**

**Comment 3:** Area A(t) (and Aice and Aliq) is still not properly defined. You simply state that it is the area of the ice particle, melted hydrometeor ...
Formulations like "A(t) is the area of each frozen hydrometeor and water droplet at time t," are ambiguous. This is ambiguous and may for example refer to the total surface area.
Regarding the heat transfer, it should be a contact area between hydrometeor and hot plate (or some sort of effective contact area).
Regarding your measurement method, it should be cross-sectional area as seen on your 2D images (Ap you call projected area).

**Response 3:** We agree with the reviewer and have updated the area definition in the manuscript. $A_p$ is defined as a contact area of the droplet associated with conductive heat transfer from the hotplate to the hydrometeor in the vertical direction. We have added this in Lines 85-90. At any time during melting and evaporation, the temperatures of some portions of the hydrometeor area are less than freezing and others greater than freezing. The contact area of a hydrometeor on the hotplate with a temperature less than or equal to zero is $A_{ice}$ (the sum of all pixels with temperatures less than or equal to zero). The contact area of a hydrometeor on the hotplate with a temperature greater than zero is $A_{liq}$ (the sum of all pixels with a temperature greater than zero). We may write A(t) = $A_{ice}$(t) + $A_{liq}$(t). We count pixels to obtain $A_{ice}$ and $A_{liq}$ using different temperature

(a) $T(t) > 0$ $\quad$ $T(t) < 0$
Hotplate temperature
without particle $T_p \sim 0$

(b) $T\,[-40,0]^oC \quad A_{ice}(t), T_{ice}(t)$ (c) $T\,(0,85]^oC \quad A_{liq}(t), T_{liq}(t)$

ranges as detailed in Figure R1 and also added in Appendix E.

**Figure R1.** Schematic illustrating the measurement area (contact area) and temperature of ice and liquid separately. (a) Thermal image of the frozen hydrometeor on the hotplate. The temperatures less than or greater than zero show the frozen hydrometeor's unmelted and melted portions, respectively. The temperature recorded by the thermal camera for the hotplate is around zero due to the low emissivity of the plate, where the actual temperature is 85 °C. (b) In post-processing the data, the temperature range was set [-40,0] °C, which allows one to "see" only the area of the unmelted portion (temperature less than and equal to zero) of the frozen hydrometeor, and the area of the melted portion (temperature greater than zero) is zero.

$T_{ice}(t)$ is the mean temperature of all pixels with a temperature less than or equal to zero, and those pixels have a good contrast with the background which allows them to be easily counted. (c) In post-processing the data, the temperature range was set (0,85] °C, allowing us to see only the area of the melted portion (temperature greater than zero) of the hydrometeor. $T_{liq}(t)$ is the mean temperature of all pixels with a temperature greater than zero, and those pixels have a good contrast with the background that allows them to count.

**Errors in Eq. (1)-(3):**

**Comment 4:** In Eq. (1), it looks like Th(t) is the temperature of the whole hydrometeor (having a cross-sectional area A(t)).
This is a simplification hiding details that you likely consider in your algorithms. In addition to the integral over time, there should be an integral over the area and Th(t), as well as Tice(t) and Tliq(t), is evaluated at each location. Presumably in your algorithm you do this double integral as two sums, one over all hydrometeor pixels of the image and one over all time steps (images) during melting and evaporation.

Without a double integral, I don't see how you can go from Eq. (1) to Eq. (2) only using A(t) = Aice(t)+Aliq(t). (In your algorithm, a pixel is either ice or liquid, thus you can split the area integral in the two parts you indicate in Eq (2). However, just as Eq. (1), Eq. (2) is wrong (over-simplified).

**Response 4:** We agree that certain details were omitted and as a result have added Eqs. (1) to (6), where double integration over time and area is explained step-by-step.

[revised manuscript text omitted]

**Comment 5:** To say that the camera doesn't see ice, and therefore Aice=0 is wrong. If starting with ice, then of course neither is Aice zero, nor the terms m*Cice*(T0 -Tice)+m*Lf. You seem to assume that heat transfer through Aice goes exclusively into temperature increase of ice and melting of ice (and heat transfer through Aliq does not go into these ice terms). So I would clearly state this assumption (important since during part of the time integral in Eq. (3) ice is still present) and directly introduce Eq. (3), i.e. skipping Equations (1) and (2), making sure the integral is properly formulated. It is worth already

here mentioning that when the camera is set to only see hydrometeors after melting, this is done in post- processing only.

**Response 5:** We agree with the reviewer regarding the confusing sentence "the camera doesn't see ice" as neither $A_{\mathrm{ice}}$ is zero, nor the expression m*C$_{\mathrm{ice}}$*(T$_0$ -T$_{\mathrm{ice}}$)+m*Lf. First, we assume that heat transfer through the area $A_{\mathrm{ice}}$ exclusively increases the temperature of the ice leading to ice melting, and that heat transfer through $A_{\mathrm{liq}}$ does not go into these ice portions (added on Lines 100-105). We justify this since the temperature gradients between the ice and water are much smaller than the temperature gradients between the plate and hydrometeor. Furthermore, the thermal conductivity of aluminum is much higher than ice or water. Note that in post-processing data, the thermal camera can be adjusted selectively to "see" particles on the hotplate in specific temperature ranges (see section 3) as detailed in Appendix E. "In post-processing data, the temperature range of the thermal image was set such that only one phase, either ice or liquid, could be seen on the hotplate. Figure E1a shows a grey thermal image where the temperature of some portions is less than or equal to zero, and some are greater than zero. In post-processing data, only

the ice portion is visible when the temperature range [-40,0] °C is used, as seen in Fig. E1b. The sum of all visible areas is $A_{ice}(t)$, and the spatial mean temperature over all those pixels is called $T_{ice}(t)$. Similarly, when the temperature range (0,85) °C is used, only the liquid portion is visible, as seen in Fig. E1c. The sum of all visible areas is $A_{liq}(t)$, and the spatial mean temperature over all those pixels is called $T_{liq}(t)$.".

**Comment 6:** Be consistent, e.g, if using Aliq, then use it also in Eq. (3).
Reformulate "Note that the initial and final temperature of all frozen hydrometeors is T0 = 0 degC andTp, respectively, during evaporation", which is confusing as you refer to a "frozen hydrometeor" (during part of the evaporation part of the hydrometeor is frozen and below T0).
In Eq. (3) it is wrong to use Cliq*Tp. The specific heat Cliq needs a temperature difference (i.e. Tp-T0). The same applies to Equations (B1) and (B2).

**Response 6:** This has been corrected. We have added the following on Lines [109-111]:
Note that all liquid droplet's initial and final temperature during evaporation is $T_0 = 0$ °C and $T_p$, respectively. In Eq. (3), the temperature difference is $T_p$-$T_0$, equal to $T_p$. $T_0$= 0 for all liquid droplets.

**Sect 2.2 Particle density**
**Unclarities remain about h and volume estimate:**

**Comment 7:** Fig. 1 is not "illustrating" well a heat transfer rate and control volume. They need to be better explained in the text and equations.
That m*Lff is "the sum of the internal energy per unit mass of a frozen hydrometeor and its latent heat of fusion" is another example of a wrong/sloppy formulation.

**Response 7:** Fig. 1 has been updated with a side and top view of the control volume and heat transfer balance across the control volume. $mL_{ff}$ is the sum of internal energy and its latent heat of fusion (energy received by ice to increase temperature and melt completely). We also added Lines 130-135.

**Comment 8:** What is the reason or motivation behind hypothesis Eq. (5)?
Explaining DeltaTmelt below Eq. (5), you have Tp(t). But earlier you stated that Tp was constant. You refer to the wrong Appendix (should be B not A).
DeltaTevap is not defined.Equation (8) is wrong.

**Response 8:** In Eq. (5) (now it is Eq. (9)), $v_{melt}$ is associated with conductive heat flux from the plate to the ice particle, which is a function of $\Delta T_{melt}$. We corrected $T_p(t)$ to $T_p$. Also corrected is Appendix B. $\Delta T_{evap} = \overline{T_p - T_{liq}(t)}$ temporal and spatial mean during evaporation is added in lines 145-150 and Eq. (8) (now it is Eq. (13)) has been corrected.

**Comment 9:** Without your explanation in the response, one needs some guessing to understand the reasoning behind the "simple height relationships like h = 2R/3". R is (only in the caption of Fig. 1 introduced as "radius of the hydrometeor", h as the "effective thickness of the hemisphere". You should (more clearly) refer to the special case of a hemispherical ice particle here. You still need a general definition of h in the text. This is important for clarity of course, but also as the definition of vmelt seems to be based on h. It is unclear how, in the laboratory, you can use Eq. (9) with its term hij. How can you determine all hij?

"...used Eq. (9) to calibrate laboratory ice particles and compare snowflake habits." What do these refer to?

Line 137: "Note that are impacted by variability..." Something is missing.

In caption of Fig. 2 "(a) Time series of the area of the ice particle (dashed black line) and the melted portion of the ice particle (solid black line)" is another example of a wrong/sloppy formulation.

**Response 9** This has been added in the text lines 150-155: "Snowflakes of complex shapes do not have simple height relationships like h = 2R/3 as shown in Fig. 1a. Hence, a method is required to determine the height and volume of every pixel within a snowflake." We added a general definition of effective thickness, $h = (1/N)\sum_i \sum_j h_{ij}$ for frozen hydrometeor on Line 135. Eq. (9) (now it is Eq. (14)) provides pixel-by-pixel volume without simplification, which gets a more accurate value and is used for calibrating laboratory ice particles and comparing snowflake habits. Eq. (10) (now it is Eq. (15)) has some simplification, as you can see in Eq. (15) , which adds around 2% additional uncertainties. Eq. (15) was used for bulk (many particles) measurements that save computational time. Line 137 (now it is Line165) is corrected. Fig. 2(a) caption is updated as a time series of the area of the ice particle, $A_{ice}(t)$ (dashed black line), and the liquid, $A_{liq}(t)$ (solid black line).

**Sect 2.3 bulk snowpack-derived quantities**

**Comment 10:** The assumption "... by assuming neither leaving any space between snowflakes nor overlapping" is wrongly placed, it is not needed for the average density defined by Eq. (12). The snow accumulation rate in Eq. (13) needs the above assumption. The assumption, however, seems unmotivated and Eq. (13). Link your text "Note that the bulk..." to that assumption.

**Response 10:** It has been corrected on Line 190. The average density of snowflakes (we deleted the average density of freshly fallen snowpack layer) and assumption "assuming no space between snowflakes nor overlapping snowflakes" added in Eq. (18) (previously, it was Eq. (13)).

**Sect 3.1: ice particle height vs effective height**

**Comment 11:** You don't seem to refer to Fig. 3 (only to Fig. 3d).
Fig. 3b, Fig. 3d, and in text related to Fig 3d, the height h seems to be a height rather than the effective height h. Also, in Tab. 1, h seems to be close to R derived from Ap (not to 2/3*R as I would expect) if the ice particle were a hemisphere. In Fig. 3b, the ice particle doesn't look hemispherical (max height is less than R). Deff in that figure is not explained. L. 376-377 also talk about height rather than effective height. Check for consistency.
If the side view in Fig. 3b is taken with the thermal camera shown in Fig. 3b, i.e. from above (Response 19), then I am not sure how it can be a side view.
You should be clearer and more detailed in your description of preparing ice particles in Sect.s 3 and 4 and resulting shapes and contact angles. I am wondering about the role of the silicone mold and didn't understand earlier that it has a certain shape (circular deepening or flat or other shapes deepenings as suggested by Line 315?).

**Response 11:** Figure. 3 is cited in the paper now at the beginning of section 3.1. Figure. 3b is updated. In Figure. 3b, h is replaced by R. In Table 1, the maximum height, which is the radius of a hemisphere, is corrected (h is replaced by R). we are not using Deff in this figure. Lines 376-377 are also corrected. For Fig. 3b, the thermal camera was recorded from the side of the ice particle (added in the caption of Fig. 3b.). The contact angle between silicon flat mold and water droplet is about 90$^\circ$ and a sample of water droplet on silicon flat mold and random surface is shown in Fig. F1.

**Fig. 5**

**Comment 12:** In the text related to Fig.5 explaining the geometrical volume estimates, it is unclear what Dv is (column III aggregates).

**Response 12:** We have added text on Lines 295-300 to address this. $D_{max}$, $D_{min}$, and $D_v$ are the lengths of three mutually perpendicular ellipsoid axes.

**Approximations in Appendix B**

**Comment 13:** As you equate (B1)=(B2) you need to relate Equations (B1) and (B2) to the whole mass (that is melted completely in B1 and then evaporated completely in B2). You need to specify over which part of the area A (or pixels) you efectively integrate in each of these equations (see comments on integrals in Equations (1) and (2) above). Explain what the "averaging approximation" means. Note that you already use an approximation when using an average DeltaTmelt and DeltaTevap that should be explained (or refer to somewhere in the paper). (See your Response 10, for example)

**Response 13:** We equated Eq. (B1) for complete melting and Eq. (B2) for complete evaporation for a single pixel (m changed to m$_{ij}$). Based on experimental tests, we assume that $\Delta t_{ij,melt} \approx \Delta t_{melt}$ and $\Delta t_{ij,evap} \approx \Delta t_{evap}$ ("averaging approximation") in Eq. (B4) that yields the relation Eq. (B5). All details are included in Appendix B. Please see Eqs. (1) to (6), where double integration over time and area is explained step-by-step.

**Appendix D errors:**

**Comment 14:** An error of only 1.4% for the area measurement seems very good. Can you describe briefly how this was determined or estimated?

Uncertainties in h: are these based on the laser measurements or the indirect method involving assumptions and approximations around DeltaTevap*Delta_tevap?

**Response 14:** Uncertainty in area measurement is estimated using the following equation

$$Error\ in\ A = \frac{\Delta A}{\bar{A}}\ 100$$

Uncertainties in h were reported based on uncertainty in $c_{melt}$, $\Delta T_{evap}$, and $\Delta t_{evap}$.